# Neural responses to syllable-induced P1m and social impairment in children with autism spectrum disorder and typically developing Peers

**Masuhiko Sano[1], Tetsu Hirosawa[1,2]☯\*, Yuko Yoshimura[2,3]☯, Chiaki Hasegawa[2], Kyung-Min An[4], Sanae Tanaka[2], Ken Yaoi[2], Nobushige Naitou[1], Mitsuru Kikuchi[1,2]**

1 Department of Psychiatry and Neurobiology, Graduate School of Medical Science, Kanazawa University, Kanazawa, Japan, 2 Research Center for Child Mental Development, Kanazawa University, Kanazawa, Japan, 3 Faculty of Education, Institute of Human and Social Sciences, Kanazawa University, Kanazawa, Japan, 4 School of Psychology, University of Birmingham, Birmingham, United Kingdom

☯ These authors contributed equally to this work.

\* hirosawatetsu1982@yahoo.co.jp

**Data Availability Statement:** All relevant data are within the manuscript and its Supporting Information files.

## Abstract

In previous magnetoencephalography (MEG) studies, children with autism spectrum disorder (ASD) have been shown to respond differently to speech stimuli than typically developing (TD) children. Quantitative evaluation of this difference in responsiveness may support early diagnosis and intervention for ASD. The objective of this research is to investigate the relationship between syllable-induced P1m and social impairment in children with ASD and TD children. We analyzed 49 children with ASD aged 40–92 months and age-matched 26 TD children. We evaluated their social impairment by means of the Social Responsiveness Scale (SRS) and their intelligence ability using the Kaufman Assessment Battery for Children (K-ABC). Multiple regression analysis with SRS score as the dependent variable and syllable-induced P1m latency or intensity and intelligence ability as explanatory variables revealed that SRS score was associated with syllable-induced P1m latency in the left hemisphere only in the TD group and not in the ASD group. A second finding was that increased leftward-lateralization of intensity was correlated with higher SRS scores only in the ASD group. These results provide valuable insights but also highlight the intricate nature of neural mechanisms and their relationship with autistic traits.

## Introduction

Autism spectrum disorder (ASD) is a neurodevelopmental disorder characterized by impaired social interaction and communication along with restricted and repetitive behavioral patterns and fixated interests, as defined in the Diagnostic and Statistical Manual of Mental Disorders, Fifth Edition (DSM-5) [1]. Early diagnosis and intervention are vital for optimizing outcomes in individuals with ASD [2–4]; However, clinical diagnosis of ASD in young children can be challenging, as the characteristic symptoms may be less evident during the early developmental

**Funding:** This study was supported by the Center of Innovation Program of the Japan Science and Technology Agency, JST, JSPS KAKENHI Grant Numbers 20H04993 and 19K02952. This research was partially supported by grants from the Moonshot Research and Development Program (grant number JPMJMS2297) of Japan Science and Technology. The funders had no role in study design, data collection and analysis, decision to publish, or preparation of the manuscript.

**Competing interests:** The authors have declared that no competing interests exist.

stages. Surveys of families with children affected by ASD highlight common delays between the initial emergence of caregiver concerns and the comprehensive evaluation as well as between the evaluation and official ASD diagnosis [5–7]. Notably, a recent multicenter surveillance study reported that, while 85% of caregivers noted concerns regarding developmental delays by 36 months of age, only 61% of the children underwent a comprehensive evaluation by 48 months. The median age at diagnosis was 52 months [3].

Diagnosing ASD proves to be challenging due to several factors including time constraints during office visits, the subtle nature of social developmental milestones, and the variability of signs and symptoms observed in individual children. The process can be further complicated by numerous elements that potentially delay the diagnosis, including the presence of less severe symptoms, female gender, concurrent issues such as anxiety or hyperactivity, lack of continuous care, and others such as socioeconomic factors and language barriers [7–10]. Moreover, the symptoms can be obscured or exacerbated by coexisting problems, which may affect both the timing and accuracy of the diagnosis. The lapse in establishing a timely diagnosis is clinically concerning as it might postpone the implementation of evidence-based behavioral interventions, potentially leading to suboptimal outcomes [11]. Implementing interventions such as the Early Start Denver Model, a behavioral therapy specifically designed for children with ASD, has been shown to enhance social, language, and cognitive functions, especially when initiated before the age of 5 (between 12 and 60 months) [12–14]. These findings underscore the importance of early diagnosis and intervention to improve the prognosis and quality of life of individuals with ASD. Given the fluctuating and sometimes elusive nature of behavioral autistic traits highlighted above, delving into the biological and physiological characteristics of ASD may forge a path towards more precise diagnostics and nuanced evaluations of treatment responses.

In recent years, brain imaging techniques have become primary methods for probing the neural foundations of ASD. Numerous neuroscience studies have employed tools such as magnetic resonance imaging (MRI), functional near infrared spectroscopy, positron emission tomography (PET), electroencephalography (EEG), magnetoencephalography (MEG), and transcranial magnetic stimulation. In light of this, McPartland and colleagues conducted an extensive review of the advancements in understanding ASD using these techniques [15]. They concluded that, while this body of research has offered critical insights, consistent findings across different studies remain elusive—with some exceptions, such as Kang et al. [16]. This lack of consistency might be due to a predominant emphasis on unveiling new results rather than solidifying existing knowledge, which can inadvertently overlook potentially significant findings, as demonstrated by Kang et al. [16]. Additionally, the inherent heterogeneity of ASD, which is diagnosed based solely on behavioral criteria and covers a broad spectrum of neural anomalies, necessitates an approach that acknowledges potential variations in neural pathology across individuals.

A more nuanced strategy might correlate specific aspects of autistic traits, such as the severity of social challenges or the manifestation of restricted and repetitive behaviors, with their neurological foundations. Given the early onset of ASD symptoms, it is especially beneficial to target younger demographics in these studies. However, when focusing on the use of imaging techniques in young children, we encounter certain limitations. For instance, it is challenging to use MRI methodologies, including functional MRI and diffusion tensor imaging, with young children. The primary obstacles are children's sensitivity to noise and the need for them to remain motionless during scans. The use of PET imaging adds to these challenges because of the introduction of radioactive tracers, which pose significant safety concerns.

Both MEG and EEG stand out as safer alternatives. They operate without noise and avoid radiation exposure risks, making them safe, noninvasive, and direct methods for measuring

the brain's magneto-electrical activity. These techniques yield detailed data that include frequency and phase information, enabling a deeper understanding of neural activity during information processing, even without evident behavior [17]. Importantly, MEG exhibits less sensitivity to conductivity variations among different anatomical structures, like the brain, cerebrospinal fluid, skull, and scalp, compared to EEG. This is because MEG measures magnetic fields rather than electric potentials [18,19]. Given these advantages, MEG holds significant promise for ASD research, especially in pediatric populations.

Auditory-evoked potentials (AEPs in EEG recordings and AEFs in MEG recordings) represent the auditory system's electromagnetic signals, generated in response to sound stimuli. These signals, precisely captured through EEG or MEG, are distinguished based on their occurrence time post-stimulus onset: early responses within 10 ms, middle-latency responses between 10 and 50 ms, and long-latency responses between 60 and 500 ms [20]. The long-latency response, especially notable in the cortical region, can be outlined using averaging techniques to enhance the target response's signal-to-noise ratio (SNR) [21–23].

Central to our discussion is the long-latency response of AEF, characterized by three notable peaks at approximately 50 ms (P1m), 100 ms (N1m), and 200 ms (P2m). Within these, the first and second peaks are often the focus of examination. However, it is important to note that, in children younger than 10 years old, the second peak may not be fully developed and might be less discernible, making the first peak a more reliable measure to assess the auditory cortex response in this age group [24–27]. In early childhood, the latency of these peaks deviates significantly from adult patterns, resulting in varied nomenclatures for the first peak across different studies, including P1m [28,29], M50 [25,30,31], P50m [32], and P100m [33]. In this study, we will adopt the term P1m for consistency. The P1m, primarily generated by neural activity in the primary and associative auditory cortices [34], acts as a pre-attentional response, reflecting the developmental status of the central auditory pathways. Specifically, its amplitude and latency indicate neural synchrony and auditory stimulation transmission time, respectively [34–36].

In numerous studies, a consistent observation is the delayed latency in the P1m component of AEFs elicited by pure tones in individuals with ASD. Roberts et al. [37] noted this in children with ASD (average age: 10.41 ± 2.51 years) compared to TD children (average age: 10.88 ± 2.70 years). This observation was confirmed by Matsuzaki et al. [38], who extended the research to include both children and adults with ASD (children: 10.07 ± 2.38 years, adults: 23.80 ± 6.26 years) and their TD counterparts (children: 9.21 ± 1.60 years, adults: 26.97 ± 1.29 years). Further studies consolidated these findings, linking longer P1m latencies with poorer language and communication skills in children with ASD ranging in age from 8 to 12 years [39]. This body of research is supported by works from Stephen et al. [40] (child participants aged 22.5 ± 2.6 months and 40.6 ± 2.5 months for TD and ASD groups respectively) and Demopoulos et al. [41] (child participants aged 11.47 ± 3.48 years and 13.78 ± 3.57 years for TD and ASD groups respectively), leading to a consensus that atypical auditory cortex neural activity is a significant characteristic of ASD, manifesting as prolonged pure-tone-evoked P1m latencies when compared to TD controls. Despite these findings, it is noteworthy that a recent meta-analysis by Williams et al. found no practically significant group differences in P1m intensities, adding a nuanced perspective to the ongoing discourse [42]. Overall, these studies suggest a complex picture of atypical auditory cortex neural activity in individuals with ASD, primarily manifesting as prolonged pure-tone-evoked P1m latencies compared to TD controls, though intensity differences remain inconclusive.

In our preceding research, we shifted the focus to syllable-induced P1m, specifically employing the Japanese syllable /ne/ as an auditory stimulus, which is rich in prosodic information and social cues [29,43–47]. This choice of stimulus, inherently not purely auditory,

potentially mirrors the aberrant processing of social information in children with ASD. Yoshimura et al. reported that, among 59 TD children with an average age of 48.6 ± 8.5 months, a weak intensity of syllable-evoked P1m in the left hemisphere was associated with lower skills in conceptual inference [29]. The conceptual inference was gauged using the riddle subscale of the K-ABC [48]. In a subsequent study, Yoshimura et al. [45] compared 33 TD children and 30 children with ASD within roughly the same age range (TD children aged 67.4 ± 10.7 months, children with ASD aged 66.9 ± 12.0 months). For TD children, a shorter latency of syllable-induced P1m in either hemisphere was related to higher skills in conceptual inference. Notably, these correlations were not significant in children with ASD.

Highlighting the intricate nature of the relationship, Yoshimura et al. observed that the association between characteristics of P1m (i.e., latency and intensity) and skills in conceptual inference differed depending on the stimuli. In a study using pure-tone instead of the human voice to induce P1m, 46 TD children (aged 70.3 ± 5.9 months) and 29 children with ASD (aged 74.7 ± 10.8 months) were examined. The results indicated that neither the latency nor intensity of pure-tone-induced P1m in either hemisphere correlated with conceptual inference in TD children. In contrast, among the ASD group, a shorter latency in the left hemisphere was linked to enhanced conceptual inference skills [49].

Yoshimura et al. continued their investigations by studying the relationship between the evolution of conceptual inference skills and changes in P1m latency and intensity over time. They engaged 20 TD children and conducted two measurements. The participants' ages were 51.0 ± 9.7 months at the first measurement and 69.0 ± 8.9 months at the second measurement. A significant increase in the intensity of P1m in the left hemisphere was strongly correlated with better development of conceptual inference skills. However, the latency of syllable-evoked P1m showed no significant relation to this development [50].

In this context, Kikuchi et al. [51] ventured to compare conceptual inference skills in children with ASD (aged 71.3 (62–92) months) and TD children (aged 70.8 (60–82) months). (This study did not provide standard deviations for the age data.) The researchers identified that children with ASD exhibited significantly lower conceptual inference skills, suggesting that diminished skills in conceptual inference could be an aspect of autistic symptomatology. Given the observed connection between syllable-evoked P1m and conceptual inference skills in TD children, it is intriguing to consider if syllable-evoked P1m might also relate to other facets of autistic symptomatology. However, the specifics of how syllable-evoked P1m interplays with the severity of autistic symptoms remain uninvestigated.

Here, we explicitly acknowledge the exploratory nature of the present study. Accordingly, our hypotheses are formulated on provisional grounds: (i) Stronger intensity of the syllable-evoked P1m in the left hemisphere corresponds with better conceptual inference skills among TD children [11], (ii) diminished conceptual inference skills potentially reflect certain facets of autistic symptomatology [51], and (iii) TD children typically display a leftward lateralization in syllable-induced P1m, which is characterized by a more pronounced intensity in the left hemisphere compared to the right. Additionally, this lateralization seems to be subdued in children with ASD [29,45]. Given these observations, we postulate that a reduced intensity of syllable-evoked P1m, especially in the left hemisphere, correlates with more pronounced autistic traits. Furthermore, a decreased leftward lateralization in the intensity of this response—potentially indicated by a diminished intensity in the left hemisphere coupled with an augmented intensity in the right—also signifies more accentuated autistic traits. To validate our hypothesis, we propose employing linear regression models to predict the degree of autistic traits, as denoted by scores on the SRS [52], using the data derived from the intensity measurements of the syllable-evoked P1m in both hemispheres. Additionally, we aim to assess any correlation between P1m latency and the severity of autistic traits. The knowledge gleaned from

this investigation holds promise for substantially influencing the clinical approach towards diagnosing and managing ASD. By identifying a more objective and noninvasive metric for autistic traits, our ambition is to set the stage for more timely diagnosis and early interventions.**Method**

## Experimental design and sample size calculation

In this study, we evaluated the severity of autism symptoms in child participants, both with and without ASD, using the SRS [52]. We assessed their intelligence using the Japanese version of the K-ABC [48]. The syllable-evoked P1m data were derived from MEG recordings. Our primary goal was to investigate the relationship between autistic symptoms, as indicated by the total T-scores on the SRS, and the intensity of the P1m response across children with and without ASD. To capture a comprehensive view of this relationship, we employed a multiple linear regression model. This model considered the possible influence of fluid intelligence, as measured by the Mental Processing Scale (MPS) from the K-ABC, on autism symptoms [53].

Specifically, our model aimed to predict the total T-scores of the SRS based on the (log-transformed) intensity of P1m from either the left or right hemisphere and the MPS scores in the K-ABC. To determine the required sample size for this investigation, we began by estimating the effect size using a squared multiple correlation coefficient ($R^2$) based on a preliminary sample [54]. This sample comprised data from six TD children from our prior studies [29,45]. Our preliminary analysis, conducted on this sample, produced $R^2$ values of 0.365 and 0.469 for models considering the right and left P1m (log-transformed) intensities, respectively. To be conservative, we chose to proceed with the smaller $R^2$ value of 0.365. Setting the alpha at 0.05 and the power (1—beta) at 0.80, we arrived at an effect size $F^2$ of 0.574 [55], determining a total sample size of 21 to accommodate the two predictors. We used G*Power version 3.121.6 [56,57] for this sample size computation. We concluded to enlist at least 25 participants in each group. This margin would accommodate potential exclusions for any unforeseen reasons.

## Participants

We recruited participants from Kanazawa University and affiliated hospitals, securing 57 children with ASD and 26 TD children. The diagnosis of ASD followed the criteria in the Diagnostic and Statistical Manual of Mental Disorders, Fourth Edition (DSM-IV) [58], utilizing either the Diagnostic Interview for Social and Communication Disorders (DISCO) [59] or the Autism Diagnostic Observation Schedule (ADOS) [60]. To mitigate the potential confounding effects of intellectual disability, we excluded six children with ASD who scored below 70 on the K-ABC Mental Processing Scale. Additionally, two children with ASD were excluded due to missing data concerning head location during the MEG recording. Consequently, our study included 49 children with ASD (37 boys and 12 girls; aged 40–92 months) and 26 TD children (21 boys and 5 girls; aged 42–89 months). Table 1 presents the characteristics of the participants.

The Research Center for Child Mental Development at Kanazawa University (https://kodomokokoro.w3.kanazawa-u.ac.jp/en/) continuously recruits children both with ASD and TD children as part of the research initiative known as the "Bambi Plan," which focuses on ASD research. Our participant pool was drawn from individuals recruited at this center between the years 2009 and 2014. We accessed their data between September 1 and September 30, 2022, for research purpose and had access to information that could identify individual participants during or after data collection. Notably, there was an overlap in the participant pool with some of our previous studies [29,45–47,51]. We integrated all available data from these earlier studies, which included 8 TD children (7 boys and 1 girl, aged 42–75 months) and

**Table 1. Characteristics of participants.**

| N | ASD N = 49 | TD N = 26 | t or χ² | p |
|---|---|---|---|---|
| Sex (%boys) | 76% | 80% | 0.268 | 0.605 |
| Age in months | 66.8 (10.9) | 65.8 (12.9) | −0.324 | 0.747 |
| SRST-scores | | | | |
| Total | 72.1 (11.9) | 49.2 (6.5) | −9.097 | 0.000* |
| Social awareness | 65.9 (9.9) | 48.0 (7.4) | −8.132 | 0.000* |
| Social cognition | 73.4 (11.7) | 50.2 (9.6) | −8.694 | 0.000* |
| Social communication | 69.7 (12.2) | 48.5 (6.3) | −8.273 | 0.000* |
| Social motivation | 60.9 (10.6) | 52.4 (7.9) | −3.572 | 0.001* |
| Autistic mannerism | 74.8 (15.5) | 48.6 (7.6) | −8.099 | 0.000* |
| K-ABC Mental processing scale score | 101.8 (15.1) | 101.4 (11.2) | −0.117 | 0.908 |

Numbers are mean (standard deviation) or counts.

K-ABC, Kaufman Assessment Battery for Children.

*$p < .05$.

21 children with ASD (19 boys and 2 girls, aged 40–92 months), supplementing it with new participants. While there was an overlap in the data, the focal points and results of the current study are distinct from those of previous research. Exclusion criteria were defined, ruling out potential participants with (1) blindness, (2) deafness, (3) any other neuropsychiatric disorder, or (4) an ongoing medication regimen. Written informed consent was obtained from parents of the children prior to their participation in the study. The Ethics Committee of Kanazawa University Hospital approved the methods and procedures, all of which were conducted in accordance with the Declaration of Helsinki.

## Psychological assessment

We used the SRS to assess the participants' autistic traits. The SRS is a 65-item rating scale used to quantify sociality and autistic mannerisms in both TD children and children with autism spectrum conditions. It generates a single measure by assessing social awareness, social cognition, social communication, social motivation, and autistic mannerisms. The SRS was completed by the parents of each participant in both groups, and we utilized gender-normed T scores (referred to as SRS-T) for each subscale in our analyses. Higher SRS-T scores indicate more severe autistic traits. While research on the validity of children's self-ratings is ongoing, the SRS can be administerd by a parent, teacher, or other adult informant. The SRS is rated based on the observation of children in their natural social contexts, reflecting what has been observed over weeks or even months rather than a single clinical or laboratory observation [52]. This feature of the SRS enables it to leverage the informant's accumulated knowledge of the child's behavior over time. Researches has shown that the SRS shows good agreement with other parent- or teacher-reported assesments of ASD related behaviors, such as the Social Communication Questionnaire [61–63], Children's Communication Checklist [63,64], and Social and Communication Disorders Checklist [65]. Additionally, the SRS scores are known to exhibit high inter-rater reliability [52] and are distributed continuously throughout a population [66].

For this study, we evaluated the intelligence of the participants using the Japanese version of the K-ABC. The K-ABC is a widely used standardized test designed to distinguish intelligence from knowledge [48,67]. The K-ABC defines a set of problem-solving skills as intelligence and includes a range of subtests designed to assess various aspects of intelligence, such

as short- and long-term memory, fluid ability, language development, reasoning, and verbal and non-verbal comprehension. Combining these subtests, intelligence is measured using the Mental Processing Scale [48]. In our study, we administered the K-ABC Mental Processing Scale score to the children in both groups (ASD and TD), and their scores were presented as standardized scores that were age-adjusted, normalized to have an average of 100, and a standard deviation of 15. The K-ABC is a well-established measure of intelligence and has demonstrated good reliability and validity across the age range of 2.5 to 12.5 years [48,67].

## MEG recordings

The MEG data were recorded with a 151-channel superconducting quantum interference device (SQUID) and a whole-head coaxial gradiometer MEG system for children (PQ 1151R; Yokogawa⁄KIT, Kanazawa, Japan) in a magnetically shielded room (Daido Steel, Nagoya, Japan) installed at the MEG Center of Ricoh Company Limited (Kanazawa, Japan). We used a customized child-seized MEG. The child MEG system ensures that the sensors can be effectively positioned within the reach of the child's brain and that the head movement is well constrained [68]. To encourage the children and minimize movement during the measurements, an experimenter was present in the room. During the MEG recordings, auditory stimuli were presented, as described below. The children were instructed to watch a silent video on a screen while listening to the auditory stimuli. MEG recordings were conducted for 12 minutes during the presentation of stimuli, and bandpass-filtered MEG data (0.16–200 Hz) were collected at a sampling rate of 1000 Hz. During MEG recordings of the children, we employed three coils to create a magnetic field associated with distinct brain landmarks (both mastoid processes and nasion) in order to track their head placement. Anatomical data from MRI could not be obtained because of the children's sensitivity to noise and difficulty remaining still during the scans.

## AEF stimuli and procedures

In this study, we employed a typical oddball paradigm in which sequences comprised standard stimuli 83% of the time (456 times) and deviant stimuli 17% of the time (90 times). The standard stimulus maintained a stable pitch contour throughout the pronunciation of the syllable /ne/, while the deviant stimulus featured a falling pitch. These sounds were articulated by a female native Japanese speaker and captured using a condenser microphone (NT1-A; Rode, Silverwater, NSW, Australia). As depicted in Fig 1, each stimulus persisted for a duration of 342 ms. The consonant /n/ lasted for 65 ms, preceding the vowel /e/. The time between stimuli was 818 ms at a level of around 65 dB (A-weighted), compared to an average background noise of 43 dB, as determined by an integrating sound level meter (LY20; Yokogawa, Tokyo, Japan). Participants received the auditory stimuli binaurally, meaning through both ears. The stimuli were transmitted via loudspeakers (HK195 Speakers; Harman Kardon, Stamford, CT, USA) located outside the magnetically shielded room housing the MEG equipment. The speakers delivered the sound into the MEG chamber through a specialized sound-conduction system that utilized a gap or aperture in the chamber's structure, ensuring the sound quality was maintained without interfering with the MEG's magnetic field. This setup facilitated a 12-minute stimulus-presentation session.

 This figure presents the sound waveforms of the standard /ne/ (left panel) and deviant /Ne/ (right panel) voice stimuli used in the study. The total duration of each stimulus is 342 ms, segmented into 65 ms for the consonant /n/ and 277 ms for the subsequent vowel sound /e/. This illustration is intended to provide a clear understanding of the structural and temporal characteristics of the stimuli employed in our experiments. The MEG analysis onset time was defined

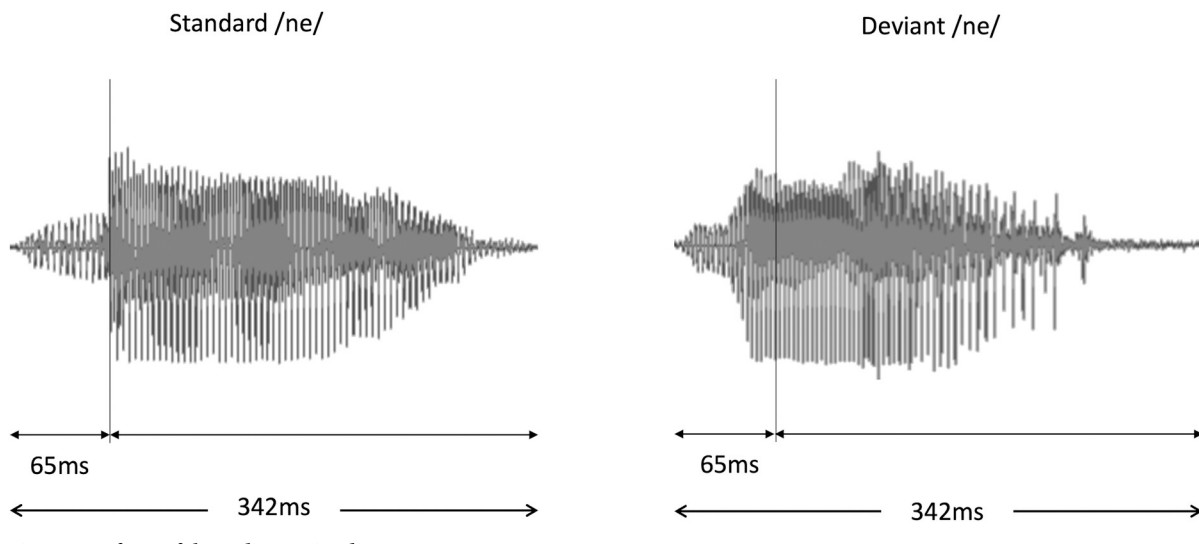

**Fig 1. Waveform of the auditory stimuli.**

as the beginning of the vowel portion. It is important to note that only the standard stimuli were used for the subsequent equivalent current dipole (ECD) estimation, as only this condition provided a sufficient number of epochs for accurate ECD calculation.

## Equivalent current dipoles for the AEFs

To identify the source of the P1m component of the AEF, we used a widely accepted method called ECDs. MEG signals are thought to originate from the apical dendrites of the pyramidal cells in the cerebral cortex. The ECD model is an idealized point source of neural activation, which represents coherent activation of a large number of pyramidal cells in a small area of the cortex. In this study, we used ECDs to estimate the source of the P1m component of the AEF.

To calculate the ECDs without using MRI-based anatomical information, we adapted a spherical model to represent the volume conductor, positioning it at the center of the MEG helmet. The onset time of the syllable stimuli (designated as 0 ms) was set at the onset of the vowel /e/ rather than the consonant /n/ in accordance with our prior studies [29,45–47,51]. We then averaged the time series spanning from -150 ms to 1000 ms (with a minimum of 300 epochs for standard stimuli) for every sensor, after baseline correction.

The baseline was set from −50 ms to 0 ms relative to the onset of the vowel /e/. Artifact-contaminated segments (such as eye blinks, eye movements, and bodily motions, typically exceeding ± 4 pT) were omitted from the analysis. A uniform ECD model facilitated the computation of the current sources, engaging at least 42 sensors per hemisphere [80]. All procedural steps and parameters were aligned with our earlier studies [29,45–47,50,51], ensuring the findings from this research can be directly compared with our prior results.

We acknowledge the significance of selecting the appropriate onset time for syllable stimuli, specifically between the consonant /n/ and the vowel /e/. To maintain consistency with our previous studies [29,45–47,51], which predominantly examined the vowel /e/ response, we opted for this latter setting in our current study. This choice is pivotal for enabling direct comparisons with prior findings, thereby enriching our understanding of auditory processing in typically developing children and children with ASD. However, we must acknowledge a crucial assumption in this approach: a minimal brain response to the consonant /n/ due to its lower sound intensity compared to the vowel /e/. While this assumption would be reasonable,

it may not fully encapsulate the natural auditory processing mechanisms and could inadvertently obscure the brain's response to the consonant /n/. This aspect warrants careful consideration in interpreting our findings.

We approved the estimated ECDs if the following conditions were met: (i) the goodness of fit (GOF) surpassed 85% during the response's target period, which indicates how well the estimated ECD matches the measured MEG signal; (ii) the position of the estimated dipoles in the single ECD model remained steady within a ± 5 mm range for each coordinate for a minimum of 6 ms; (iii) dipole intensities were equal to or less than 80 nA; and (iv) the direction of the estimated ECD was in an anterosuperior orientation. The time point was identified as the latency, when the estimated dipole intensity value had reached a maximum and met the above criteria within the time window from 40 to 150 ms. Importantly, only the standard stimuli were used for the subsequent ECD estimation, as only this condition provided a sufficient number of epochs for accurate ECD calculation after artifact rejection across all child participants.

## Statistical analysis

Our primary hypothesis was that a reduced intensity of syllable-evoked P1m, especially in the left hemisphere, correlates with more pronounced autistic traits. Additionally, a decreased leftward lateralization in the intensity of this response also signifies more pronounced autistic traits.

First, to evaluate this hypothesis, our statistical model aimed to explore any potential differential correlation between the intensity of syllable-evoked P1m and autistic traits between children with ASD and TD children. Recognizing the potential influence of intellectual abilities on SRS scores, as suggested in prior research [69], we incorporated this variable into our model. Thus, we performed a linear regression analysis predicting the SRS total T-score based on the intensity of the left (or right) P1m, diagnostic category (ASD or TD), MPS scores, and an interaction term between P1m intensity and diagnosis. This interaction term is vital for discerning potential variations in the relationship between P1m intensity in the respective hemispheres and autistic traits among the two groups. The inclusion of MPS scores mitigates the potential confounding effect of intelligence on SRS scores. We conducted this regression twice—once for the left hemisphere and once for the right—without adjusting for multiple comparisons, given the few preplanned and likely correlated comparisons [70,71]. Consequently, we set the statistical significance level at $p < .05$. Aligning with our previous methods [29,45], we used log-transformed intensity values rather than raw intensity for comparisons. For thoroughness, we also executed the same analysis using P1m latency in place of log-intensity.

Second, to delve deeper into the relationship between syllable-induced P1m intensity and autistic traits, we sought to discern how changes in the P1m intensity, either diminished in the left hemisphere or amplified in the right, might correlate with more pronounced autistic traits within each diagnostic group (i.e., TD children and children with ASD). We performed separate linear regression analyses for each group predicting the SRS total T-score based on P1m intensity (either left or right) and MPS scores. This analysis was undertaken four times—once for each hemisphere within both diagnostic categories—without corrections for multiple comparisons, setting the significance threshold at $p < .05$, given the preplanned and potentially correlated comparisons [70,71]. For completeness, P1m latency was also examined in lieu of log-intensity.

Third, we sought to identify any varying correlations between the degree of leftward lateralization in P1m intensity and autistic traits across children with ASD and TD children. A linear regression analysis was undertaken to predict the SRS total T-score based on the P1m's

leftward lateralization (defined as log-transformed P1m intensity in the left hemisphere minus its counterpart in the right), diagnostic category, MPS scores, and an interaction between P1m leftward lateralization and diagnosis. Similarly, we conducted an analysis replacing log-intensity with P1m latency with a significance level set at p < .05, following the rationale of limited preplanned and potentially interlinked comparisons [70,71].

Fourth, we aimed to probe the association between the leftward lateralization of syllable-induced P1m and autistic traits within each diagnostic group. We performed separate linear regression analyses for each group predicting the SRS total T-score based on P1m's leftward lateralization and MPS scores. This analysis phase was executed twice, once for each diagnostic category, again without multiple comparison corrections. The statistical significance threshold was retained at p < .05 due to the limited preplanned and potentially correlated comparisons [70,71]. As a thoroughness measure, we also examined P1m latency as an alternative to log-intensity.

Before applying any linear regression model, we verified that our data meet the assumptions for regression analysis. Specifically, we used standard methods to verify the linearity, normality, homogeneity of variance, influence, and collinearity. As a result, the assumption of homogeneity was violated in some of the regression models. Therefore, we used heteroscedasticity-robust standard errors [72].

## Results

MEG data were collected from 26 TD children and 49 children with ASD. Only responses to the standard stimuli yielded a sufficient number of epochs for accurate ECD calculation across participants. Therefore, only responses to standard stimuli were analyzed. Eleven participants did not achieve the minimum of 300 epochs. Specifically, data from two TD children and nine children with ASD were excluded. The numbers of participants excluded were not significantly different between groups ($\chi 2$ = 1.55, p = 0.214). Thus, the responses were averaged over 403.8 ± 2.2 (mean ± SD) epochs for the 24 TD children and 382.4 ± 39.1 epochs for the 40 children with ASD. A Student's t-test revealed that the average number of epochs was significantly higher for the TD children (t(62) = 2.26, p = 0.03). The syllable-induced AEF displayed prominent peaks in the 40–150 ms time window (i.e., P1m) for the majority of participants in both hemispheres when the baseline was set from −50 ms to 0 ms relative to the onset of the vowel /e/. Fig 2 presents these waveforms and the magnetic contour map of P1m for a representative participant. S1 Fig displays the group averages of waveforms for the TD and ASD participants in the left and right hemispheres.

To ensure that the initial head positions did not differ statistically between the ASD and TD groups, we attached three coils to each participant's skull, positioned at both mastoid processes and the nasion. Each coil created a magnetic field that enabled us to track their initial head positions. A Student's t-test revealed significant differences between the ASD and TD groups in the y-coordinate of the coil at both the left mastoid process (t(62) = -2.22, p = 0.03) and the right mastoid process (t(62) = -2.05, p = 0.04), which might affect the results (as discussed in the limitations section). No significant differences were observed in the x and z coordinates of these coils. Similarly, no significant differences were found in any coordinate of the coil at the nasion. Detailed results are presented in the S1 Table.

This figure presents these waveforms and the magnetic contour map of P1m for a representative participant. Syllable-induced AEF with a baseline from −50 to 0 ms relative to the onset of the vowel /e/. The resultant AEF displayed a pronounced activity peak between 45 and 150 ms. The onset of the consonant /n/ is at −65 ms relative to that of /e/. The blue arrow displays the direction of the estimated dipole moment.

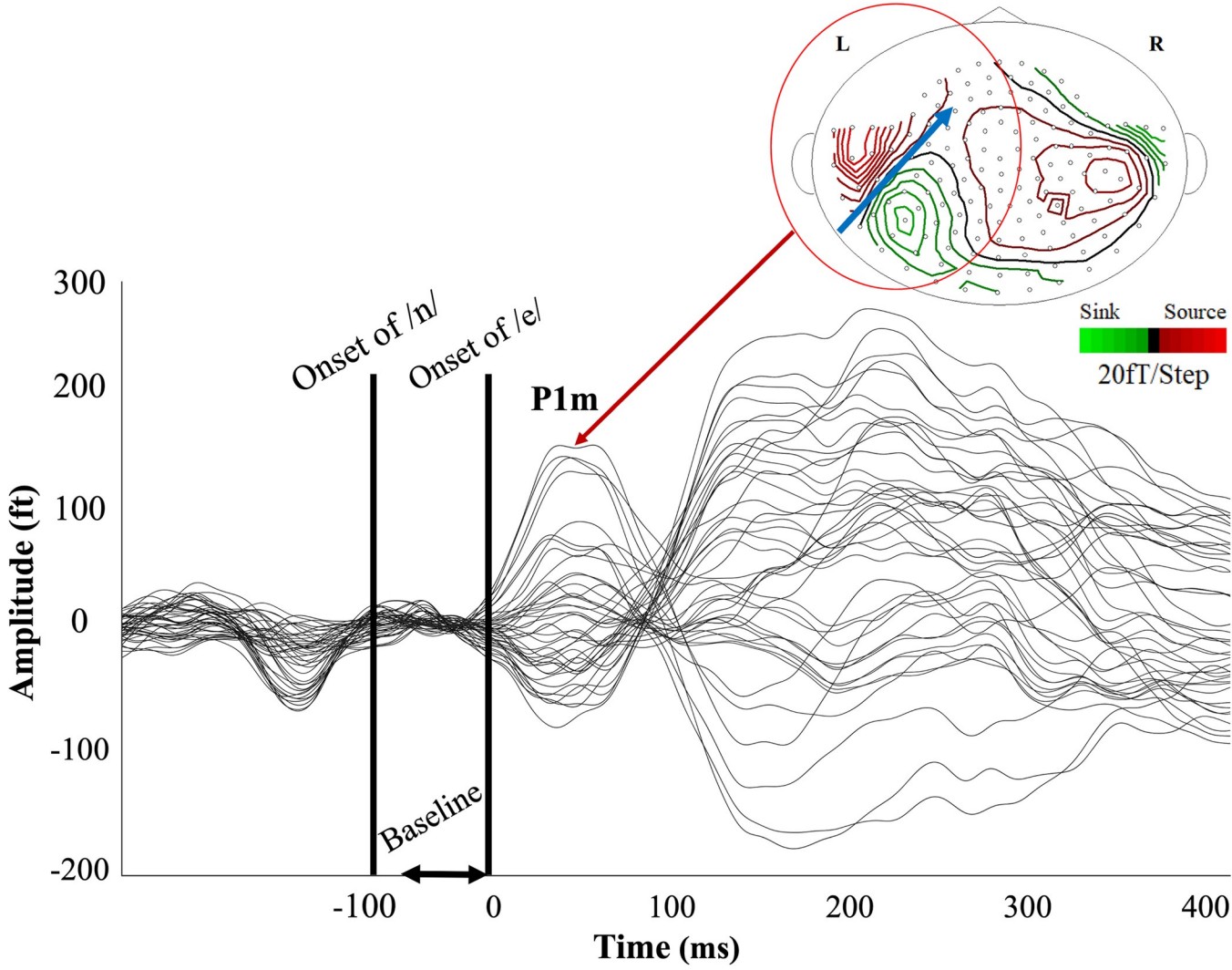

**Fig 2. Neuromagnetic response to the standard syllable /ne/ stimuli.**

In total, the ECD sources for P1m in the left hemisphere were reliably estimated for 21 TD children and 38 children with ASD. The dipole latency in the left hemisphere was 78.7 ± 19.2 (mean ± SD) ms for TD children and 78.4 ± 20.1 ms for children with ASD. The difference in latency between the two groups was not significant (t(57) = 0.06, p = 0.96). The log-transformed intensity of these dipoles was 2.8 ± 0.5 for TD children and 2.7 ± 0.6 for children with ASD. The difference in log-transformed intensity between the two groups was also not significant (t(57) = 0.71, p = 0.48).

For the right hemisphere, the ECD sources for P1m were reliably estimated for 23 TD children and 33 children with ASD. The dipole latency was 72.7 ± 20.1 (mean ± SD) ms for TD children and 81.9 ± 18.7 ms for children with ASD. The difference in latency between the two groups was not significant (t(54) = −1.74, p = 0.09). The log-transformed intensity of these dipoles was 2.6 ± 0.4 for TD children and 2.7 ± 0.4 for children with ASD. Again, the difference in intensity was not statistically significant between the two groups (t(54) = −0.84, p = 0.41). Fig 3 presents violin plots of P1m latency and log-transformed intensity within each group.

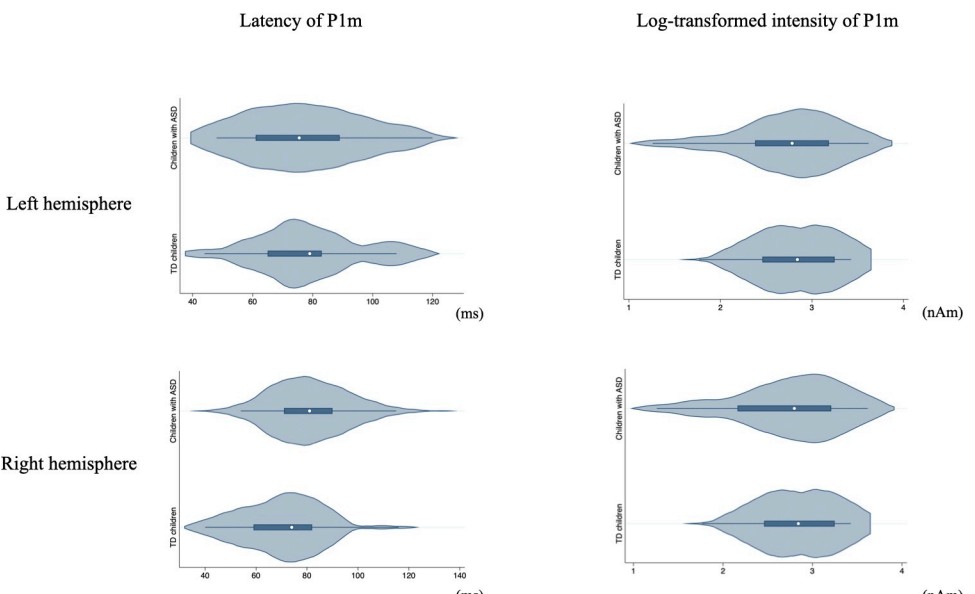

**Fig 3. Violin plots of P1m latency and log-transformed intensity by diagnostic group.**

Besides reporting no significant group differences in P1 latency and log-transformed intensity, we also examined the asymmetry of P1 intensity between the TD and ASD groups. It is important to note that this analysis revealed no significant difference in the asymmetry of P1 amplitude between the two groups. We delve into the details of this analysis, including the calculation method and considerations for outlier exclusion, in a later section of this manuscript.

The figure displays violin plots illustrating the distribution of P1m latency and log-transformed intensity values for both the right (R) and left (L) hemispheres, separated by diagnostic group (ASD and TD). The top two plots represent the right hemisphere with the first showcasing P1m latency and the second depicting the log-transformed P1m intensity. The bottom two plots correspond to the left hemisphere; the first illustrates P1m latency, and the second demonstrates the log-transformed P1m intensity. The width of each "violin" indicates the density of the data at different values, offering a visual representation of the data's distribution.

TD children, typically developing children; ASD, Autism spectrum disorder.

## Significant relationship between left P1m latency and SRS T-scores

We conducted linear regression analyses to predict the SRS total T-score with variables including left (or right) P1m intensity, diagnostic group (ASD or TD), MPS scores, and an interaction term between left (or right) P1m intensity and diagnosis. Consistent with our previous methods [29,45] and as described in the Method section, we utilized log-transformed intensity values rather than raw intensities for these models. These models did not yield any statistically significant factors. Separate regression analyses were then performed for each group to predict the SRS total T-score based on left (or right) P1m log-intensity and MPS scores. Again, no significant factors emerged from these models. Details are provided in S2 and S3 Tables.

Transitioning our focus from intensity to latency, we conducted linear regression analyses with variables including left (or right) P1m latency, diagnostic group, MPS scores, and the corresponding interaction term. Notably, only the effect of left P1m latency was significant (t(54) = −2.64, p = 0.011). Additional separate regressions for each diagnostic group were then performed using left (or right) P1m latency and MPS scores. Within these, only the TD group

**Table 2. Association between SRS total T-score and right or left P1m latency controlling for K-ABC mental processing scale score.**

| | Coeff. | Robust SE | t | p | 95%CI | | F | Prob > F | $R^2$ |
|---|---|---|---|---|---|---|---|---|---|
| **vs. SRS total T-score** | | | | | | | | | |
| Right P1m Latency | −0.11 | 0.84 | −1.37 | 0.177 | −0.28 | 0.05 | 21.42 | <0.001 | 0.55 |
| Diagnosis | 9.23 | 10.26 | 0.90 | 0.372 | −11.37 | 29.83 | | | |
| Interaction between right P1m latency and diagnosis | 0.15 | 0.13 | 1.16 | 0.253 | −0.11 | 0.41 | | | |
| Mental processing scale score | 0.06 | 0.11 | 0.56 | 0.579 | −0.15 | 0.27 | | | |
| **vs. SRS total T-score** | | | | | | | | | |
| Left P1m latency | −0.14 | 0.06 | −2.64 | 0.011* | −0.29 | −0.04 | 28.82 | <0.001 | 0.54 |
| Diagnosis | 14.64 | 9.77 | 1.50 | 0.140 | −4.95 | 34.23 | | | |
| Interaction between left P1m latency and diagnosis | 0.08 | 0.12 | 0.70 | 0.486 | −0.15 | 0.32 | | | |
| Mental processing scale score | −0.00 | 0.12 | −0.04 | 0.971 | −0.24 | 0.23 | | | |

Coeff., regression coefficient; SE, standard error; CI, confidence interval.

*p < .05.

showed a significant effect for left P1m latency (t(18) = −2.59, p = 0.018). The results from these analyses are presented in Tables 2 and 3. Fig 4 shows a visual representation of the relation between left P1m latency and SRS total T-score.

To visualize the relation between SRS total T-scores and P1m latency in the left hemisphere for TD children, we performed a simple regression to predict SRS total T-scores based solely on P1m latency, excluding the mental processing scale for clarity. The effect of P1m latency on SRS total T-scores remains significant in this simplified model (t(19) = -2.68, p = 0.015). The figure depicts individual data points for TD children. The solid line represents the predicted regression line, and the shaded area around it denotes the 95% confidence intervals based on our regression model.

SRS, social responsiveness scale; TD children, typically developing children.

**Table 3. Association between SRS total T-score and right or left P1m latency for each diagnosis group controlling for K-ABC mental processing scale score.**

| | Coeff. | Robust SE | t | p | 95%CI | | F | Prob > F | $R^2$ |
|---|---|---|---|---|---|---|---|---|---|
| **vs. SRS total T-score** | | | | | | | | | |
| **TD** | | | | | | | | | |
| Right P1m latency | −0.12 | 0.09 | −1.37 | 0.187 | −0.3 | 0.06 | 1.07 | 0.36 | 0.14 |
| Mental processing scale score | −0.04 | 0.08 | −0.46 | 0.651 | −0.21 | 0.13 | | | |
| **ASD** | | | | | | | | | |
| Right P1m latency | 0.04 | 0.1 | 0.39 | 0.702 | −0.17 | 0.24 | 0.31 | 0.73 | 0.02 |
| Mental processing scale score | 0.1 | 0.15 | 0.71 | 0.485 | −0.19 | 0.41 | | | |
| **vs. SRS total T-score** | | | | | | | | | |
| **TD** | | | | | | | | | |
| Left P1m latency | −0.17 | 0.65 | −2.59 | 0.018* | −0.3 | −0.03 | 3.76 | 0.04 | 0.22 |
| Mental processing scale score | −0.07 | 0.11 | −0.62 | 0.541 | −0.29 | 0.16 | | | |
| **ASD** | | | | | | | | | |
| Left P1m latency | −0.08 | 0.1 | −0.8 | 0.427 | −0.28 | 0.12 | 0.33 | 0.72 | 0.02 |
| Mental processing scale score | 0.01 | 0.14 | 0.09 | 0.93 | −0.28 | 0.3 | | | |

Coeff., regression coefficient; SE, standard error; CI, confidence interval.

*p < .05.

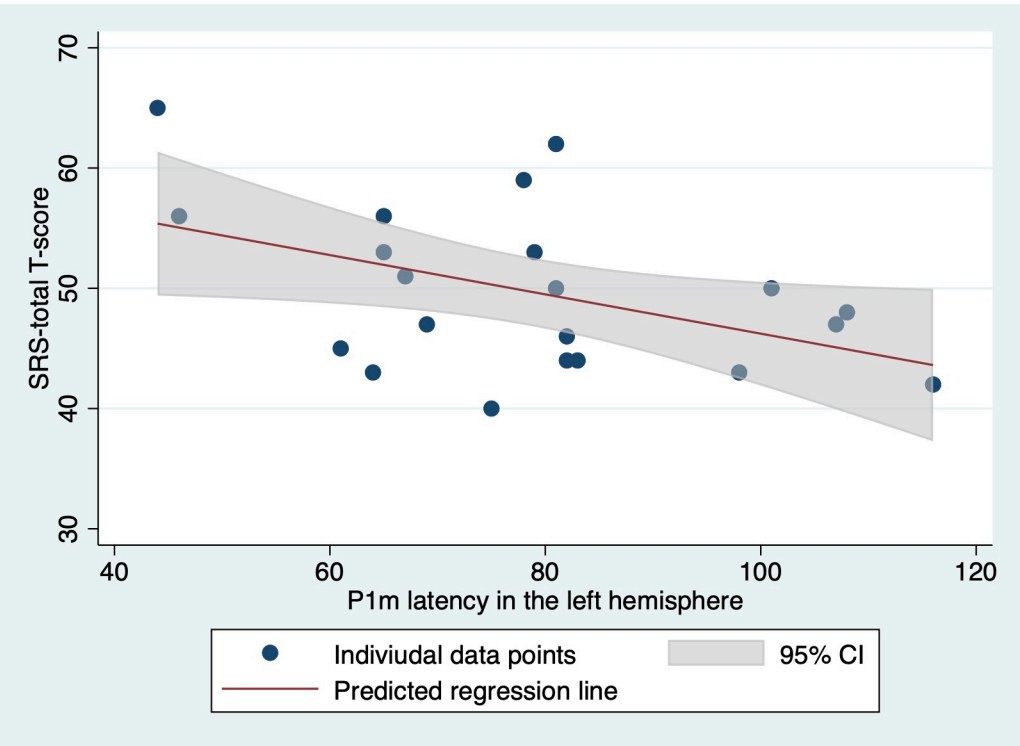

**Fig 4. Relationship between SRS Total T-Scores and P1m latency in the left hemisphere.**

Given the significant difference in the average number of epochs between the ASD and TD groups, we introduced another regression model to account for the SNR. We used the square root of the number of averages for each participant as a proxy for SNR [73] and added it to the original regression model. In particular, this revised model predicts the SRS total T-score based on the left P1m latency, diagnostic group (ASD or TD), MPS scores, interaction between left P1m latency and diagnosis, and square root of the number of averages. By integrating the SNR proxy, we ensure relationships observed with left P1m latency are not only a result of SNR variations. Even with this adjustment, the left P1m latency remained a significant predictor (t(53) = −2.62, p = 0.011). Adopting a similar approach for the separate group-based regressions, the left P1m latency remained a significant predictor (t(17) = −2.61, p = 0.018) within the TD group. All other factors remained nonsignificant. Detailed results are provided in S4 Table.

In summary, our findings emphasize a significant association between more pronounced autistic traits and the shorter latency of syllable-induced P1m, predominantly in the TD group and localized to the left hemisphere. Such a relationship was not evident with P1m log-intensity.

## More pronounced autistic symptoms are associated with stronger leftward lateralization in P1m intensity, exclusively in children with ASD

We performed a linear regression analysis to predict the SRS total T-score using P1m's left-ward lateralization (defined as the log-transformed P1m intensity in the left hemisphere minus its counterpart in the right), diagnostic group, MPS scores, and an interaction term between P1m leftward lateralization and diagnosis. For this analysis, we only included participants for

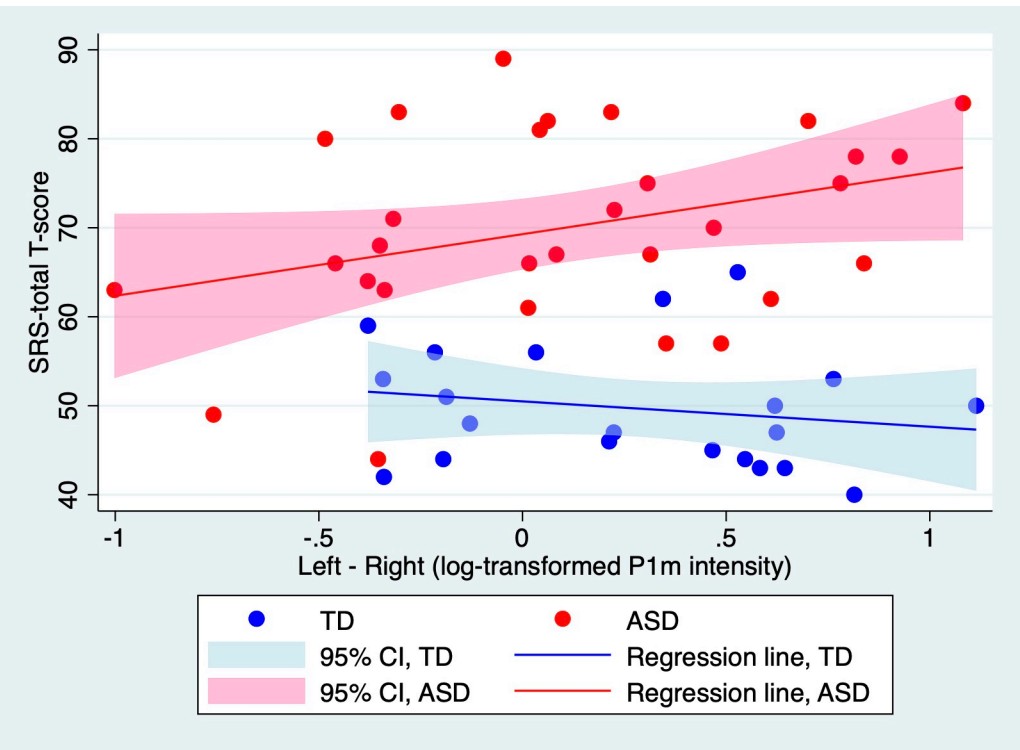

**Fig 5. Relationship between SRS Total T-Scores and P1m's leftward lateralization.**

whom an ECD could be reliably estimated in both hemispheres. Prior to executing this model, we identified a potential outlier; one child with ASD exhibited extremely low leftward lateralization (evidenced by a z-score below −3). Grubb's test confirmed this observation as the sole outlier in our sample. Consequently, we excluded this participant from our analysis. This resulted in participants comprising 21 TD children and 30 children with ASD with leftward lateralization of 0.27 ± 0.45 nAm for TD children and 0.06 ± 0.61 nAm (mean ± SD) for children with ASD. The difference in leftward lateralization between the two groups was not significant (t(50) = 1.36, p = 0.18). Within this model, both the interaction effect (t(46) = 2.20, p = 0.033) and the effect of diagnosis (t(46) = 7.27, p < 0.001) were found to be significant. Fig 5 provides a visual representation of this interaction. We then conducted separate regressions for each group, considering leftward lateralization and MPS scores. Notably, only in the ASD group was the effect of leftward lateralization significant (t(27) = 2.32, p = 0.028). Results from these analyses are provided in Table 4.

As significant differences were observed in the initial head position (i.e., y-coordinate of the coils at both the left and right mastoid process), we investigated whether the leftward lateralization in log-transformed P1m intensity correlated with these initial head positions. To this end, we employed simple regression analysis to predict the leftward lateralization in log-transformed P1m intensity based on the x, y, or z coordinates of each coil separately. A significant correlation was found between the leftward lateralization in log-transformed P1m intensity and the x coordinate of the coil at the nasion (t(50) = -2.61, p = 0.01), indicating that a larger leftward lateralization of intensity corresponds to a left-located coil at the nasion. No significant associations were observed in any of the other models. Detailed results are presented in the S5 Table.

**Table 4. Association between SRS total T-score and leftward lateralization in P1m log-intensity controlling for K-ABC mental processing scale score.**

| | Coeff. | Robust SE | t | $p$ | 95%CI | | F | Prob > F | $R^2$ |
|---|---|---|---|---|---|---|---|---|---|
| **vs. SRS total T-score** | | | | | | | | | |
| Leftward lateralization in log-intensity | −2.70 | 3.09 | −0.87 | 0.387 | −8.92 | 3.52 | 22.56 | <0.001 | 0.59 |
| Diagnosis | 18.68 | 2.57 | 7.27 | <0.001* | 13.51 | 23.85 | | | |
| Interaction between leftward lateralization in log-intensity and diagnosis | 9.81 | 4.46 | 2.20 | 0.033* | 0.84 | 18.78 | | | |
| Mental processing scale score | 0.05 | 0.11 | 0.43 | 0.667 | −0.18 | 0.28 | | | |
| **vs. SRS total T-score** | | | | | | | | | |
| **TD** | | | | | | | | | |
| Leftward lateralization in log-intensity | −2.95 | 3.27 | −0.9 | 0.379 | −9.81 | 3.92 | 0.41 | 0.67 | 0.04 |
| Mental processing scale score | −0.03 | 0.12 | −0.28 | 0.782 | −0.28 | 0.21 | | | |
| **ASD** | | | | | | | | | |
| Leftward lateralization in log-intensity | 7.22 | 3.12 | 2.32 | 0.028* | 0.83 | 13.62 | 3.18 | 0.06 | 0.13 |
| Mental processing scale score | 0.08 | 0.15 | 0.54 | 0.595 | −0.23 | 0.39 | | | |

Coeff., regression coefficient; SE, standard error; CI, confidence interval.

Leftward lateralization in log-intensity is defined as the log-transformed P1m intensity in the left hemisphere minus its counterpart in the right.

*p < .05.

This figure illustrates the relationship between SRS total T-scores and P1m's leftward lateralization in log-transformed intensity (defined as the log-transformed P1m intensity in the left hemisphere minus that in the right) for ASD and TD children. Separate simple regressions were performed for each group to predict SRS total T-scores based on this measure of P1m's leftward lateralization, excluding the mental processing scale for clarity. The effect of leftward lateralization in log-transformed P1m intensity on SRS total T-scores was found to be significant only for TD children (t(28) = 2.15, p = 0.04). Individual data points for each group are plotted, with each line corresponding to a diagnostic group, illustrating how predicted SRS scores vary with P1m's leftward lateralization.

SRS, social responsiveness scale; ASD, autism spectrum disorder; TD children, typically developing children.

Using the approach previously described and recognizing the significant difference in epoch averages between the ASD and TD groups, we adjusted our model to factor in the SNR, using the square root of averages as an SNR proxy [86]. Post-adjustment, the interaction between leftward lateralization and diagnosis retained its significance (t(45) = −2.38, p = 0.022). Similarly, in the adjusted regression for the ASD group, the effect of leftward lateralization remained significant (t(26) = 2.81, p = 0.009). Comprehensive results of this modified analysis are available in S6 Table.

In a subsequent step, we modified our regression analyses by replacing the leftward lateralization intensity with its latency counterpart (defined as P1m latency in the left hemisphere minus that in the right) to predict the SRS total T-scores. The latency-based leftward lateralization was 4.95 ± 18.24 (mean ± SD) ms for TD children and −1.94 ± 21.86 ms for children with ASD. The difference between the two groups was not significant (t(50) = 1.18, p = 0.24). For this latency-based leftward lateralization, no potential outliers were observed. This modification yielded no significant predictors with the exception of the effect of diagnosis (t(47) = 8.28, p < 0.001). These details are found in S7 Table.

In summary, our findings demonstrate that pronounced autistic symptoms correlate with enhanced leftward lateralization of P1m intensity exclusively in children diagnosed with ASD. This relationship is absent in TD children. Moreover, a similar correlation was not observed when considering leftward lateralization in terms of latency.

### Results of analyses with new participants only

In the present study, some participants overlapped with participants included in our previous studies. We excluded them and performed all analyses on 'new' participants only to test whether the results could be reproduced. Twenty-eight children with ASD and 18 TD children were included in these analyses. To summarize the main results, the relationship between left P1m latency and SRS T-scores in the TD group did not maintain statistical significance; the association between SRS total T-score and leftward lateralization in P1m log-intensity in the ASD group f remained statistically significant. Other detailed results are given in the S8 Table.

## Discussion

The primary objective of this study was to investigate the relationship between syllable-induced P1m responses and the severity of autistic traits. We postulated that a reduced intensity of the P1m response, especially in the left hemisphere, would be indicative of more pronounced autistic traits. Additionally, we hypothesized that a decrease in leftward lateralization of the response's intensity—potentially characterized by a diminished intensity in the left hemisphere accompanied by an enhanced intensity in the right—would also signify heightened autistic traits. Beyond this, we sought to determine any potential correlation between P1m latency and the severity of autistic traits.

Our empirical observations, however, present a nuanced understanding of our hypotheses: (i) Contrary to our initial supposition, the most notable association between autistic traits and the syllable-induced P1m response was observed in terms of latency rather than intensity. This relationship was predominantly evident in the TD group with a significant effect localized in the left hemisphere. Such a correlation with the P1m log-intensity was absent. (ii) Our second key observation was that, in children diagnosed with ASD, more pronounced autistic symptoms were significantly associated with an increased leftward lateralization of the P1m intensity. Interestingly, this correlation was not seen in the TD group. Additionally, when evaluating leftward lateralization in terms of latency, no such relationship was discerned.

We identified a significant association between a shorter latency of syllable-induced P1m in the left hemisphere and pronounced autistic traits. Interestingly, this correlation is primarily evident in TD children and appears nonsignificant in children with ASD. At first glance, this finding suggests a potential link between neural auditory processing and autistic traits. However, it is crucial to consider that the SRS scale may not reflect the same underlying physiology in TD and ASD individuals. In TD children, the SRS could represent a range of cognitive processing styles related to autistic traits, while in ASD children, it might reflect more specific aspects of ASD pathology, such as excitatory/inhibitory (E/I) imbalances [74]. From this perspective, the observed correlation between shorter P1m latency and higher SRS scores in TD children could indicate a relationship between P1m latency and a broad spectrum of cognitive processing that relates to autistic traits rather than being solely indicative of autism-specific pathology. This spectrum might include neural adaptations or processing efficiencies unrelated to autism but still captured by SRS scores. Conversely, the lack of a significant correlation in ASD children hints at the involvement of different neural processes that are reflected in their SRS scores. These processes could be linked to specific neurophysiological characteristics, such as E/I imbalances, which are considered characteristic of ASD pathology [74]. Therefore, the shorter P1m latency in TD children might reflect a neural process less directly related to autistic pathology, perhaps involved in more general social information processing. In contrast, in children with ASD, given their pronounced social deficits, such general social information processing might no longer be associated with the severity of their social deficits. Instead, their social deficits might be more directly related to autism-specific pathology, which

might not be reflected in P1m latency. This could explain why we failed to find a significant relation between P1m latency and SRS scores in this population. Our findings highlight the need for further research to explore the specific neurophysiological underpinnings of SRS scores in both TD and ASD individuals. Future studies should aim to disentangle the relationships between neural markers like P1m latency, autism-specific pathology, and other mechanisms involved in social information processing. Such research would offer a clearer understanding of the complex interplay between auditory processing and social responsiveness traits in both populations.

Further insight can be gleaned from Yoshimura et al.'s research [50]. In their study, the relationship between the evolution of conceptual inference skills and shifts in P1m latency was explored over time. By engaging TD children and taking two measurements—initially around 51 months and subsequently around 69 months—it was discerned that changes in the latency of syllable-evoked P1m were not significantly linked with the development of conceptual inference skills. Reflecting on this, if the aforementioned compensatory mechanism does exist, its influence might be more pronounced during the earlier developmental stages.

However, while the studies discussed offer intriguing insights into the potential relationship between the latency of the syllable-induced P1m in the left hemisphere and pronounced autistic traits, caution is advised in interpreting these findings. The small sample sizes inherent in these studies could introduce variability that might not be representative of the broader population. It is essential to acknowledge that these initial observations, though promising, require further validation through more extensive, population-based studies. Only with more comprehensive data can we draw definitive conclusions about the complex interplay between neural markers like P1m latency and the manifestations of autistic traits. Until then, these studies serve as a foundation, prompting deeper exploration and understanding in the realm of neurodevelopmental research.

Informed by Yoshimura et al.'s study, which noted that TD children usually display a marked leftward lateralization in syllable-induced P1m intensity—a characteristic subdued in children with ASD [45]—we initially conjectured that decreased leftward lateralization might indicate more pronounced autistic traits. Contrary to our expectations, we found a significant association between reduced leftward lateralization and milder autistic symptoms. Intriguingly, this correlation was exclusive to the ASD group and not significant in the TD group. The divergence between our initial hypothesis and the observed findings necessitates deeper reflection. One interpretation could be that the diminished leftward lateralization observed in ASD children, as seen in Yoshimura et al.'s findings [45], might be indicative of a neural compensatory mechanism. In essence, the brains of children with ASD might recalibrate, reducing their inherent leftward lateralization, to offset pronounced autistic traits.

Broadening our perspective from MEG to other imaging modalities, a consistent pattern emerges across various neuroimaging studies, all pointing towards atypical brain lateralization in ASD. For instance, Postema et al., leveraging an expansive dataset from the ENIGMA consortium, identified significant associations between ASD and alterations in cortical thickness asymmetry, especially in frontal and temporal regions [75]. Functional MRI studies also consistently report atypical lateralization across various networks in individuals with ASD [76–79]. Distinct white matter tracts in ASD further exhibit nonconventional asymmetry patterns [80–82]. Interestingly, only one study to our knowledge, by Conti et al. [83], has directly probed the correlation between the degree of brain lateralization and autistic symptoms, as measured by the ADOS-G. This study pinpointed significant associations between brain lateralization of diffusion indexes and clinical severity in newly diagnosed toddlers with ASD. Given this context, our research deepens Yoshimura et al.'s observations [45] by demonstrating a correlation between leftward P1m intensity lateralization and autistic symptoms.

Moreover, we also extend the insights from Conti et al.'s study [83]. Our findings highlight that atypical lateralization, not only in structural (DTI) but also in functional (EEG) terms, correlates with the severity of autistic symptoms.

It is important to note that we did not observe a significant association between the intensity of P1m in the left hemisphere and autistic traits in TD children. This finding is somewhat surprising, especially in light of the reported association between a stronger intensity of P1m in the left hemisphere and better conceptual inference skills among TD children [29]. Given the known association between ASD and diminished conceptual inference skills [51], one might anticipate a stronger correlation. This discrepancy suggests that the intensity of P1m in the left hemisphere might be specifically associated with conceptual inference in this population, but not necessarily with other facets of autistic symptomatology. Another potential explanation for our observation could be a lack of statistical power. Our sample size calculation indicates that a minimum total sample size of 21 is needed to accommodate two predictors (i.e., P1m log-intensity and MPS scores in K-ABC) to predict the SRS total T-score. Given this minimum requirement, the sample size for this analysis (predicting the SRS total T-score based on P1m log-intensity in the left hemisphere and MPS scores in K-ABC) might be on the threshold. As such, there is a possibility we might have missed this effect due to chance.

Lastly, another approach to interpreting our P1m results involves considering the broader framework of auditory processing, particularly in relation to the potential contribution of the 'sustained negative shift' of current, as described in adults [84] and in both adults and children [85]. The processing of sounds characterized by periodicity/pitch and/or formant structure, such as vowels, is associated with a greater sustained negative shift of cortical source current, known as the sustained field (SF), which persists throughout stimulus presentation. This SF, captured by MEG/EEG, is thought to reflect the activation of non-synchronized neuronal populations [86,87]. These neurons function as 'feature detectors' for perceptually salient features of complex sounds, facilitating higher-level processing [88–90]. The enhancement of MEG/EEG-measured SF occurs when stimuli are perceptually salient [91] or carry semantic meaning [92], and its magnitude varies with phonetic features, such as periodic versus non-periodic vowels [85]. Notably, SF is evident in the time range of the P1m component or even earlier, suggesting that the co-occurrence of SF with P1m might influence the contour and amplitude of the P1m. This interaction is particularly relevant in our findings, where we observed an association between autistic traits and syllable-induced P1m latency and its leftward lateralization in intencity, possibly reflecting a latent relationship between autistic traits and SF. The potential connection between autistic traits and SF is compelling, given the emerging behavioral and electrophysiological evidence of impaired attentional responses to speech in children with ASD. This might imply reduced perceptual salience of speech stimuli and atypical higher-level processing. Earlier studies indicate that children with ASD exhibit specific deficits in orienting to vowel sounds compared to simple and complex tones [93], highlighting their potentially reduced perceptual salience to speech. Moreover, these deficits may be linked to atypical higher-level processing of auditory stimuli in this population [94]. Given these considerations, future research aiming to further elucidate the complex interplay between autistic traits, SF, and properties of P1m could offer a richer understanding of the neural basis of ASD and how it is reflected in MEG/EEG measurements.

This study has several limitations. First, although social auditory stimuli were used in this study, no controlled experiments with auditory stimuli of a different nature, such as pure tones, were conducted; therefore, it is unclear whether the relationship between AEF and autistic traits found in this study is specific to social auditory stimuli or whether it is also found with other forms of auditory stimuli. Therefore, it is unclear whether social auditory stimuli are more useful than general auditory stimuli in assessing the relationship between AEF and

autistic traits. In the future, similar studies should be conducted using non-social auditory stimuli, such as pure tones, to verify whether similar results can be replicated. Another limitation arises from setting the baseline relative to the onset of the vowel /e/. Our study's focus on the vowel /e/, setting the baseline relative to its onset, inherently implies a possible oversight of the brain's response to the preceding consonant /n/. This approach, while methodologically sound for our current research objectives, may mask nuances in the brain's processing of the /n/ consonant. This limitation is particularly pertinent when considering the differential responses between TD children and children with ASD. Our findings, thus, should be interpreted with an awareness of this potential masking effect. In light of this, future research endeavors should contemplate including stimuli combinations like /ne/ and /e/ to comprehensively investigate the brain's distinct responses to consonants and vowels. Such explorations would be instrumental in deepening our understanding of auditory processing variations between TD and ASD groups, potentially leading to more nuanced insights into their auditory processing characteristics. There are also important limitations regarding the sample in this study. First, because the age of the sample was over 3 years, it is not known whether the association between AEF and autistic traits found in the present study is also found in children under 2 years of age, and the findings of the present study cannot be directly used for early diagnosis for these children. Furthermore, the sample size may have been small, limiting the detection power. Future studies should be conducted with a wider sample age range and larger sample size. Another limitation, as underscored by recent research, concerns the potential intersection of Auditory Processing Disorder (APD) symptoms within the ASD population. Studies by Sharma et al. [95] and Lunardelo et al. [96] revealed P1 amplitude abnormalities in children with APD in response to speech stimuli, specifically the /da/ sound, which is similar to the /ne/ sound utilized in our research. These findings imply that P1 irregularities may not be unique to ASD and could also signify a central auditory processing deficit characteristic of APD. Moreover, the work of James et al. [84] indicates a potentially high incidence of APD symptoms among children with ASD. This overlap suggests that the P1 abnormalities we observed in children with ASD might partially reflect a broader spectrum of auditory processing challenges extending beyond the confines of ASD. The absence of a direct evaluation of APD in our study is a notable oversight. This limitation warrants caution in attributing the P1 abnormalities solely to ASD and suggests the need for future research to disentangle the auditory processing profiles of ASD from those of APD. Undertaking such research would provide a more comprehensive understanding of the auditory processing dynamics in neurodevelopmental disorders. In this study, participants were monitored using a video camera to detect noticeable body movements. An examiner accompanied the children in the shielded room and instructed them to maintain a constant head position throughout the experiment. Instances of pronounced body movement were excluded based on noise detection. Additionally, participants who exhibited significant shifts in head position during the session were excluded due to a reduction in the GOF in the P1m dipole analysis. Despite these measures, we observed significant differences in the initial head positions between the two groups. Specifically, the positions of both the right and left mastoid processes in children with ASD were significantly more posterior compared to those in TD children. This difference could reflect variations in initial head positioning or head shape; either factor could potentially influence the results of dipole estimation. Indeed, while the position of the mastoid process did not affect the leftward lateralization of log-transformed P1m intensity, the x-coordinate of the coil at the nasion was found to significantly influence the estimation of this parameter. Furthermore, this study did not comprehensively account for the impact of fine head movements and variations in head shape, which are factors that could introduce additional variability in the neuroimaging data. Future

research should consider these aspects more thoroughly to mitigate their potential effects on data interpretation.

Another limitation pertains to the sample characteristics of our study. We did not observe a significant group difference in intensity-based leftward lateralization between the two groups, which contrasts with the findings from Yoshimura et al. [45]. This discrepancy could potentially be attributed to the smaller sample size in our study; P1m was reliably estimated in both hemispheres for 21 TD children and 30 children with ASD in our study while Yoshimura et al. had reliable P1m estimates for 30 TD children and 33 children with ASD. This difference in sample size might have influenced the findings. Regardless of the reason, the inconsistency limits the utility of this particular MEG variable as a potential neurobiomarker.

In conclusion, this study explored the complex relationship between syllable-evoked P1m responses and the severity of autistic traits. Guided by prior research, our initial hypotheses anticipated specific correlations, but our empirical observations nuanced these expectations. Specifically, the relationship between autistic traits and syllable-induced P1m was more pronounced with latency than with intensity, a connection predominantly observed in the TD group. For children with ASD, increased severity of autistic symptoms was associated with a more pronounced leftward lateralization of the P1m intensity. However, these insights come with a caveat. Given the limited sample size of this study, our findings should be viewed as preliminary. They set the stage for future research, emphasizing the need for more extensive, population-based investigations. In essence, our results provide valuable insights but also highlight the intricate nature of neural mechanisms and their relationship with autistic traits.

## Supporting information

**S1 Checklist.**
(DOCX)

**S1 Fig. The group averages of neuromagnetic response to the standard syllable /ne/ stimuli for the TD and ASD participants.**
(TIF)

**S1 Table. Position of the three coils on the heads of the participants.**
(PDF)

**S2 Table. Association between SRS total T-score and right or left P1m log-intensity controlling for K-ABC mental processing scale score.**
(PDF)

**S3 Table. Association between SRS total T-score and right or left P1m log-intensity for each diagnosis group controlling for K-ABC mental processing scale score.**
(PDF)

**S4 Table. Association between SRS total T-score and leftward lateralization in P1m log-intensity controlling for K-ABC mental processing scale score and signal noise ratio.**
(PDF)

**S5 Table. Correlation between the leftward lateralization in log-transformed P1m intensity and coil positions.**
(PDF)

**S6 Table. Association between SRS total T-score and leftward lateralization in P1m latency controlling for K-ABC mental processing scale score and signal noise ratio.**
(PDF)

**S7 Table. Association between SRS-total T-score and leftward lateralization in P1m latency controlling for Mental processing scale score in K-ABC.**
(PDF)

**S8 Table. All results of analyses with new subjects only.**
(PDF)

**S1 Data.**
(XLSX)

## Acknowledgments

The authors wish to thank all individuals who participated in this study and our colleagues for their invaluable assistance, particularly S. Kitagawa, Y. Saotome, M. Ozawa, Y. Morita, and T. Haruta. The authors would also like to extend their gratitude to Editage (www.editage.com) for English language editing.

## Author Contributions

**Conceptualization:** Masuhiko Sano, Tetsu Hirosawa.

**Data curation:** Yuko Yoshimura.

**Formal analysis:** Masuhiko Sano, Yuko Yoshimura.

**Funding acquisition:** Mitsuru Kikuchi.

**Investigation:** Chiaki Hasegawa, Kyung-Min An, Sanae Tanaka, Ken Yaoi.

**Methodology:** Tetsu Hirosawa, Yuko Yoshimura.

**Project administration:** Mitsuru Kikuchi.

**Supervision:** Tetsu Hirosawa, Nobushige Naitou, Mitsuru Kikuchi.

**Writing – original draft:** Masuhiko Sano.

**Writing – review & editing:** Tetsu Hirosawa.

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
