## [Decision Letter · Decision Letter 0]

11 Sep 2023

PONE-D-23-19599Right P1m predicts autistic traits in children with ASD.PLOS ONE

Dear Dr. Hirosawa,

Thank you for submitting your manuscript to PLOS ONE. After careful consideration, we feel that it has merit but does not fully meet PLOS ONE’s publication criteria as it currently stands. Therefore, we invite you to submit a revised version of the manuscript that addresses the points raised during the review process.

 Please respond each comment and highlight all your edits. ==============================

We look forward to receiving your revised manuscript.

Kind regards,

Thiago P. Fernandes, PhD

Academic Editor

PLOS ONE

Reviewers' comments:

Reviewer's Responses to Questions

**Comments to the Author**

1. Is the manuscript technically sound, and do the data support the conclusions?

Reviewer #1: Partly

Reviewer #2: No

2. Has the statistical analysis been performed appropriately and rigorously? 

Reviewer #1: Yes

Reviewer #2: No

3. Have the authors made all data underlying the findings in their manuscript fully available?

Reviewer #1: No

Reviewer #2: Yes

4. Is the manuscript presented in an intelligible fashion and written in standard English?

Reviewer #1: Yes

Reviewer #2: Yes

5. Review Comments to the Author

Reviewer #1: Major comments:

1. One of my major concerns is, according to Table 1 there were no significant differences in MEG right and left P1m intensities between children with and without ASD. This limited the potential use of this MEG variable as a neurobiomarker. It would be important for the authors to explicitly explain why the finding in the current paper did not replicate their previous finding and add it as a limitation.

2. According to the introduction, it seems that the P1m latency is different in children with and without ASD and could be a potential neurobiomarker. I suggest the author include P1m latency as a targeted variable.

Introduction:

1. In the introduction, the authors compared the pros and cons of multiple neuroimaging tools, such as MRI, PET, MEG, and EEG. However, functional near-infrared spectroscopy (fNIRS) is also a promising tool for studying neural activity in children with ASD. Please see McPartland et al. (2022) as an example.

Reference: McPartland, J. C., Lerner, M. D., Bhat, A., Clarkson, T., Jack, A., Koohsari, S., Matuskey, D., McQuaid, G. A., Su, W. C., & Trevisan, D. A. (2021). Looking Back at the Next 40 Years of ASD Neuroscience Research. Journal of autism and developmental disorders, 51(12), 4333–4353. https://doi.org/10.1007/s10803-021-05095-5

2. Starting from Line 68, the authors reviewed previous neural imaging findings in children with and without ASD. Since the current study is aiming for supporting early identification and intervention, it is important to include the children’s age included in the studies.

3. Since one of the implications of this study is to use MEG neurobiomarkers for early identification, it’s important to include information about when ASD symptoms are presented and the age at which children were usually diagnosed with ASD.

Methods and results:

1. Did the authors conduct any sample size estimation based on their previous study?

2. The first sentence in the result section seems to be cut out.

Discussion:

1. Please provide more rationales for the clinical implication. For example, how the findings could be used as an early biomarker, and how this neurobiomarker could be used to support early intervention.

Reviewer #2: Hirosawa and colleagues investigated in children with ASD (~3 to 7 years) and control typically developing (TD) children auditory responses to speech stimuli (syllable /ne/) using MEG. Based on their previous results the authors expected to find in children with ASD atypical amplitudes of the transient component P100 in the left and right hemispheres (increased in the right hemisphere, decreased in the left), as well as correlations of these amplitudes with autism severity estimated using SRS questionnaire.

Stimuli were presented using passive oddball paradigm. Deviant /ne/ (n=90) differed from standard /ne/ (n=456) in intonation. Duration of each stimulus was 342 ms, including 65 ms of consonant. ISI was 818 ms. The ‘zero time’ was set at the beginning of the vowel /e/. -50 to 0 ms was used as a baseline (i.e. overlapping with /n/ presentation).

Data were recorded using child-size MEG system. No individual brain models were available. Standards were used to fit the dipole model to P100, but only P100 to deviants was analyzed.

The main result is a negative correlation between P100m amplitude in the right hemisphere and SRS score in ASD.

The positive side of this study is the use of the child-size system, which allows better sensitivity to auditory magnetic fields in children. The relatively large sample of ASD participants (N=49) is also an advantage. However, I have a number of concerns.

Major issues.

1. In the Introduction and methods the authors claim that P100 may serve as a biomarker of ASD. However, abnormalities in auditory responses may be not specific to ASD, but associated with ADHD or auditory processing disorder, since both these disorders have high comorbidity with ASD.

2. The authors mentioned that part of their sample overlapped with the sample included in their other studies (e.g. Yoshimura, Mol Autism. 2013;4(1):38.), where atypical lateralization of P100 (called P50 in that study) was found. However, the authors did not specify the number of overlapping participants in each sample. Will the results be reproduced when only ‘new’ subjects are analyzed?

3. The argumentation used in the manuscript is fully based on the correlational approach, without any reflection about putative neural mechanisms underlying ASD vs TD differenses in P100, its lateralization and correlation with SRS. I doubt that this correlational approach can supply reliable biomarker (also see point 1).

4. It is unclear why standards were used to fit the dipole, but only responses to deviants were analyzed. If the authors expect differences in P100 only for deviants, they need to explain why.

5. Number of averages per subject is not specified. Is this number differs in ASD and control groups? Is the correlation still significant when SNR is taken into account, e.g. by correcting for square root of number of averages?

6. Please, present more visual information. This would be of great help for the reader to understand/interpret the results. I would be interested to see:

1. waveforms of the standard and deviant stimuli,

2. timecourses of the responses to standards and deviants (including essentially long baseline period before /n/),

3. violin plots of P100 latensies in both groups,

4. violin plots of P100 amplitudes in both groups

7. I wonder why interval -50 to 0 ms relative to the onset of /e/ vowel was chosen as the baseline? There is a risk that this baseline interval itself differs in ASD and control group. Will the result still keep if the baseline is taken before the /n/ consonant?

8. Was the head position continuously tracked during the experiment and corrected to a common position? Was the predominant head position different in ASD and control groups? For example, could differences in P100 amplitude (increased on the right and decreased in the left in ASD) be explained by differences in the head position between the groups?

9. Statistical analysis. It is unclear why the authors predict SRS with P100 amplitude, MRS and diagnosis. The meaning of significant interaction between P100 and Diagnosis is unclear.

Why not to perform ANOVA with factors Group and Hemisphere, as the authors did in their previous study (Yoshimura, 2013 MolA)?

10. lines. 319-322 ‘We further analyzed the association between right P1m intensity and the SRS-subscales and found a significant negative correlation with the SRS autistic mannerism T-score, suggesting that the relationship between P1m intensity and ASD symptoms may be primarily driven by autistic mannerisms.’

To make this conclusion the authors need to insure that correlation with mannerism subscale is significantly different from correlations with other subscales.

Minor issues

Fig.1. ‘Margins are statistics that are calculated based on a dataset where some or all of the covariates are held at fixed values different from their true values (49, 50).’

The Legend is not very helpful, as the reader needs to address citations 49, 50 to understand the plot. Please explain meaning of these margins more clearly.

Considering violation of homogeneity of variance, another approach would be to calculate partial non-parametric correlations, controlling for MRS/IQ.

Please report high-pass filter, even if only an inbuilt filter was applied and no additional filtration was used.

Line 260: Sentence starts with small ‘i’.

Abbreviation MRS is mot explained in the text.

Line 201-202: ‘Participants received the stimulus through both ears via a gap in the MEG chamber, which was transmitted by loudspeakers…’

This is unclear, please explain.’

Lines 341-343: ‘Furthermore, in contrast to the ASD group, we did not find a significant correlation between the right P1m intensity and SRS total T-score in TD children, possibly due to a lack of power in our sample (53).’

Why the authors expected to find such correlation?

The study would benefit from help of English editing services.

e.g. in abstract:

‘On performing multiple regression analyses using P1m intensity in the right and left hemispheres and the K-ABC Mental Processing Scale score as the dependent variables, and using the SRS total T-score as the independent variable, we identified right P1m intensity as a predictor of the SRS total T-score in children with ASD, and this relationship was not found in TD children. ’

6. PLOS authors have the option to publish the peer review history of their article (what does this mean?). If published, this will include your full peer review and any attached files.

Reviewer #1: **Yes: **Wan-Chun Su

Reviewer #2: No

---

## [Author Response · Author response to Decision Letter 0]

22 Oct 2023

Dear Reviewers,

　Thank you for your kind comments. In responding to your comments, we found that in our first draft, we made a methodological error in our analysis. After correcting this and performing the analysis again, the results of the analysis were different from the first draft, and the title of the first draft no longer corresponds to the correct results. We reported this to the editor and received permission to change the title.

　The following are responses to all comments.

PONE-D-23-19599

Right P1m predicts autistic traits in children with ASD.

PLOS ONE

Dear Dr. Hirosawa,

Thank you for submitting your manuscript to PLOS ONE. After careful consideration, we feel that it has merit but does not fully meet PLOS ONE’s publication criteria as it currently stands. Therefore, we invite you to submit a revised version of the manuscript that addresses the points raised during the review process.

Please respond each comment and highlight all your edits.

We look forward to receiving your revised manuscript.

Kind regards,

Thiago P. Fernandes, PhD

Academic Editor

PLOS ONE

Thank you for pointing this out. We have followed your instructions and rewritten the manuscript according to the style requirements. 

Thank you for your careful review of our manuscript and for bringing up the discrepancy between the Funding Information and Financial Disclosure sections. We understand the importance of accurate grant information for transparency and verification.

Upon checking our original submission, we found that the grant numbers we provided in the Financial Disclosure section are indeed accurate. Specifically, our study was supported by the Center of Innovation Program of the Japan Science and Technology Agency and KAKENHI with Grant Numbers 20H04993 and 19K02952.

For verification purposes, the respective grants can be accessed at the following official KAKENHI webpages:

https://kaken.nii.ac.jp/ja/grant/KAKENHI-PUBLICLY-20H04993

https://kaken.nii.ac.jp/ja/grant/KAKENHI-PROJECT-19K02952/

We appreciate the diligence in ensuring the accuracy of the funding sources. We kindly ask that you cross-check the provided links and grant numbers to confirm the correctness.

Thank you for pointing out the requirement to include captions for our Supporting Information files. We have taken care to ensure that the captions for our Supporting Information are clear and provide the necessary context for readers.

As per your suggestion, we have included the following captions for our Supporting Information at the end of our manuscript:

(Supporting information)

S1 Table. Association between SRS total T-score and right or left P1m log-intensity controlling for K-ABC mental processing scale score. 

S2 Table. Association between SRS total T-score and right or left P1m log-intensity for each diagnosis group controlling for K-ABC mental processing scale score.

S3 Table. Association between SRS total T-score and right or left P1m latency controlling for K-ABC mental processing scale score and signal noise ratio.

S4 Table. Association between SRS total T-score and leftward lateralization in P1m log-intensity controlling for K-ABC mental processing scale score and signal noise ratio.

S5 Table. Association between SRS total T-score and leftward lateralization in P1m latency controlling for K-ABC mental processing scale score and signal noise ratio. 

S6 Tables for Reviewer 2. All results with the new participants only

We have also ensured that all in-text citations to these tables are correctly referenced to match the above captions.

We appreciate your guidance and feedback, ensuring that our manuscript adheres to the PLoS ONE's Supporting Information guidelines. Please let us know if there are any further modifications required.

Review Comments to the Author

Reviewer #1: Major comments:

1. One of my major concerns is, according to Table 1 there were no significant differences in MEG right and left P1m intensities between children with and without ASD. This limited the potential use of this MEG variable as a neurobiomarker. It would be important for the authors to explicitly explain why the finding in the current paper did not replicate their previous finding and add it as a limitation.

Thank you for highlighting the discrepancy between our current results and our previous findings. We recognize the importance of addressing inconsistencies between studies, especially when considering the potential of certain variables as neurobiomarkers. In light of your comments, we explicitly acknowledge the exploratory nature of this study rather than seeking a neurobiomarker of ASD. 

(Introduction) 

Here, we explicitly acknowledge the exploratory nature of the present study. Accordingly, our hypotheses are formulated on provisional grounds: (i) Stronger intensity of the syllable-evoked P1m in the left hemisphere corresponds with better conceptual inference skills among TD children [11]; (ii) diminished conceptual inference skills potentially reflect certain facets of autistic symptomatology [62]; and (iii) TD children typically display a leftward lateralization in syllable-induced P1m, which is characterized by a more pronounced intensity in the left hemisphere compared to the right. Additionally, this lateralization seems to be subdued in children with ASD [29, 56].

Lack of Group Difference in Intensity-based Leftward Lateralization: We acknowledge this limitation in our discussion. The lack of a significant group difference contrasts with findings from Yoshimura et al. [56]. We speculate that the discrepancy could stem from the smaller sample size of our study relative to theirs. While it is tempting to look at MEG variables as potential neurobiomarkers, this study underscores the complexity involved and the caution required when making such determinations.

(Discussion)

Another limitation pertains to the sample characteristics of our study. We did not observe a significant group difference in intensity-based leftward lateralization between the two groups, which contrasts with the findings from Yoshimura et al. [56]. This discrepancy could potentially be attributed to the smaller sample size in our study; P1m was reliably estimated in both hemispheres for 21 TD children and 30 children with ASD in out study while Yoshimura et al. had reliable P1m estimates for 30 TD children and 33 children with ASD. This difference in sample size might have influenced the findings. Regardless of the reason, the inconsistency limits the utility of this particular MEG variable as a potential neurobiomarker.

We appreciate the emphasis on clarity, consistency, and a balanced interpretation of our results. Especially in our discussion, we have attempted to reflect these in our revised manuscript, focusing not on asserting definitive claims but rather on interpreting our findings within the context of existing research while also acknowledging the limitations and exploratory nature of our study. 

2. According to the introduction, it seems that the P1m latency is different in children with and without ASD and could be a potential neurobiomarker. I suggest the author include P1m latency as a targeted variable.

Thank you for your insightful comment regarding the inclusion of P1m latency as a targeted variable. We genuinely appreciate your guidance, which we believe has significantly enhanced the depth and clarity of our study.

In response to your feedback, we have revised the Introduction, Method, Results, and Discussion sections.

Revision of the Introduction: We have undertaken a comprehensive revision of the introduction to elucidate the results concerning P1m latency from previous studies. We aimed to provide a clearer context for how the latency differences in children with and without ASD could be potentially informative as a neurobiomarker.

Modifications in Methods and Results: Based on the augmented introduction and the importance of P1m latency, we have adapted our methods to target this variable as well as intensity. Consequently, our results section was also updated to reflect the findings pertaining to P1m latency.

Discussion Section Overhaul: With the inclusion of P1m latency as another focal point, it became imperative to revisit our discussion. We have rewritten a majority of this section to encompass interpretations, implications, and potential avenues for further research in light of the new results on P1m latency.

We believe that these changes not only address your concerns but also amplify the overall coherence and relevance of our manuscript. The focus on P1m latency has enriched our narrative, providing a more rounded perspective on its significance in the context of ASD.

We sincerely thank you for steering us in this direction and hope that our revisions align with your expectations.

Introduction:

1. In the introduction, the authors compared the pros and cons of multiple neuroimaging tools, such as MRI, PET, MEG, and EEG. However, functional near-infrared spectroscopy (fNIRS) is also a promising tool for studying neural activity in children with ASD. Please see McPartland et al. (2022) as an example.

Reference: McPartland, J. C., Lerner, M. D., Bhat, A., Clarkson, T., Jack, A., Koohsari, S., Matuskey, D., McQuaid, G. A., Su, W. C., & Trevisan, D. A. (2021). Looking Back at the Next 40 Years of ASD Neuroscience Research. Journal of autism and developmental disorders, 51(12), 4333–4353. https://doi.org/10.1007/s10803-021-05095-5

Thank you for drawing our attention to the inclusion of functional near-infrared spectroscopy (fNIRS) as a pertinent neuroimaging tool in the context of ASD research and for providing the reference by McPartland et al. (2022) as an illustrative example. In response to your feedback, we expanded the second paragraph of the Introduction to integrate fNIRS alongside other primary brain imaging techniques. This was done to provide a holistic overview of the array of tools employed in recent neuroscience studies focusing on ASD. In this context, we found the review by McPartland et al. extremely insightful. Accordingly, we have integrated key takeaways from their work to highlight the advancements in understanding ASD using various neuroimaging techniques as well as the challenges posed by inconsistencies in the existing literature.

(Introduction) 

In recent years, brain imaging techniques have become primary methods for probing the neural foundations of ASD. Numerous neuroscience studies have employed tools such as magnetic resonance imaging (MRI), functional near infrared spectroscopy, positron emission tomography (PET), electroencephalography (EEG), magnetoencephalography (MEG), and transcranial magnetic stimulation. In light of this, McPartland and colleagues conducted an extensive review of the advancements in understanding ASD using these techniques [15]. They concluded that while this body of research has offered critical insights, consistent findings across different studies remain elusive—with some exceptions, such as Kang et al. [16]. This lack of consistency might be due to a predominant emphasis on unveiling new results rather than solidifying existing knowledge, which can inadvertently overlook potentially significant findings, as demonstrated by Kang et al. [16]. Additionally, the inherent heterogeneity of ASD, which is diagnosed based solely on behavioral criteria and covers a broad spectrum of neural anomalies, necessitates an approach that acknowledges potential variations in neural pathology across individuals. 

A more nuanced strategy might correlate specific aspects of autistic traits, such as the severity of social challenges or the manifestation of restricted and repetitive behaviors, with their neurological foundations. Given the early onset of ASD symptoms, it is especially beneficial to target younger demographics in these studies. However, when focusing on the use of imaging techniques in young children, we encounter certain limitations. For instance, it is challenging to use MRI methodologies, including functional MRI and diffusion tensor imaging, with young children. The primary obstacles are children’s sensitivity to noise and the need for them to remain motionless during scans. The use of PET imaging adds to these challenges because of the introduction of radioactive tracers, which pose significant safety concerns. 

Both MEG and EEG stand out as safer alternatives. They operate without noise and avoid radiation exposure risks, making them safe, non-invasive, and direct methods for measuring the brain's magneto-electrical activity. These techniques yield detailed data that include frequency and phase information, enabling a deeper understanding of neural activity during information processing, even without evident behavior [17]. Importantly, MEG exhibits less sensitivity to conductivity variations among different anatomical structures, like the brain, cerebrospinal fluid, skull, and scalp, compared to EEG. This is because MEG measures magnetic fields rather than electric potentials [18, 19]. Given these advantages, MEG holds significant promise for ASD research, especially in pediatric populations.

We believe that these revisions have strengthened the introductory section of our manuscript, laying a more comprehensive foundation for the subsequent content. Your feedback has been instrumental in refining our narrative, and we are appreciative of your guidance in this regard.

2. Starting from Line 68, the authors reviewed previous neural imaging findings in children with and without ASD. Since the current study is aiming for supporting early identification and intervention, it is important to include the children’s age included in the studies.

We are grateful for your insightful comment highlighting the importance of mentioning the age of participants in the cited studies, especially given the study's emphasis on early identification of and intervention in ASD.

In response to your feedback, we made the following adjustments to the manuscript:

Age Specification: We meticulously integrated the age range of participants for each of the studies referenced. By doing so, we hope to provide the reader with a clearer context regarding the developmental stages these findings pertain to.

Age Consistency: We took care to ensure that age ranges provided were consistent with the original works, allowing for a more direct comparison between our findings and those from the referenced studies.

We believe that integrating the age information strengthens the clarity and context of our manuscript, ensuring that our readers can easily situate our findings within the developmental timeline. Your comment was pivotal in bringing about this enhancement, and we are appreciative of your keen observation.

(Introduction)

In numerous studies, a consistent observation is the delayed latency in the P1m component of AEFs elicited by pure tones in individuals with ASD. Roberts et al. [48] noted this in children with ASD (average age: 10.41 ± 2.51 years) compared to TD children (average age: 10.88 ± 2.70 years). This observation was confirmed by Matsuzaki et al. [49], who extended the research to include both children and adults with ASD (children: 10.07 ± 2.38 years, adults: 23.80 ± 6.26 years) and their TD counterparts (children: 9.21 ± 1.60 years, adults: 26.97 ± 1.29 years). Further studies consolidated these findings, linking longer P1m latencies with poorer language and communication skills in children with ASD ranging in age from 8 to 12 years [50]. This body of research is supported by works from Stephen et al. [51] (child participants aged 22.5 ± 2.6 months and 40.6 ± 2.5 months for TD and ASD groups, respectively) and Demopoulos et al. [52] (child participants aged 11.47 ± 3.48 years and 13.78 ± 3.57 years for TD and ASD groups, respectively), leading to a consensus that atypical auditory cortex neural activity is a significant characteristic of ASD, manifesting as prolonged pure-tone-evoked P1m latencies when compared to TD controls. Despite these findings, it is noteworthy that a recent meta-analysis by Williams et al. found no practically significant group differences in P1m intensities, adding a nuanced perspective to the ongoing discourse [53]. Overall, these studies suggest a complex picture of atypical auditory cortex neural activity in individuals with ASD, primarily manifesting as prolonged pure-tone-evoked P1m latencies compared to TD controls, though intensity differences remain inconclusive. 

In our preceding research, we shifted the focus to syllable-induced P1m, specifically employing the Japanese syllable /ne/ as an auditory stimulus, which is rich in prosodic information and social cues [29, 54-58]. This choice of stimulus, inherently not purely auditory, potentially mirrors the aberrant processing of social information in children with ASD. Yoshimura et al. reported that, among 59 TD children with an average age of 48.6 ± 8.5 months, a weak intensity of syllable-evoked P1m in the left hemisphere was associated with lower skills in conceptual inference [29]. The conceptual inference was gauged using the riddle subscale of the K-ABC [59]. In a subsequent study, Yoshimura et al. [56] compared 33 TD children and 30 children with ASD within roughly the same age range (TD children aged 67.4 ± 10.7 months, children with ASD aged 66.9 ± 12.0 months). For TD children, a shorter latency of syllable-evoked P1m in either hemisphere was related to higher skills in conceptual inference. Notably, these correlations were not significant in children with ASD. 

Highlighting the intricate nature of the relationship, Yoshimura et al. observed that the association between characteristics of P1m (i.e., latency and intensity) and skills in conceptual inference differed depending on the stimuli. In a study using pure-tone instead of the human voice to induce P1m, 46 TD children (aged 70.3 ± 5.9 months) and 29 children with ASD (aged 74.7 ± 10.8 months) were examined. The results indicated that neither the latency nor intensity of pure-tone-induced P1m in either hemisphere correlated with conceptual inference in TD children. In contrast, among the ASD group, a shorter latency in the left hemisphere was linked to enhanced conceptual inference skills [60]. 

Yoshimura et al. continued their investigations by studying the relationship between the evolution of conceptual inference skills and changes in P1m latency and intensity over time. They engaged 20 TD children and conducted two measurements. The participants' ages were 51.0 ± 9.7 months at the first measurement and 69.0 ± 8.9 months at the second measurement. A significant increase in the intensity of P1m in the left hemisphere was strongly correlated with better development of conceptual inference skills. However, the latency of syllable-evoked P1m showed no significant relation to this development [61]. 

In this context, Kikuchi et al. [62] ventured to compare conceptual inference skills in children with ASD (aged 71.3 (62–92) months) and TD children (aged 70.8 (60–82) months). (This study did not provide standard deviations for the age data.) The researchers identified that children with ASD exhibited significantly lower conceptual inference skills, suggesting that diminished skills in conceptual inference could be an aspect of autistic symptomatology. Given the observed connection between syllable-evoked P1m and conceptual inference skills in TD children, it is intriguing to consider if syllable-evoked P1m might also relate to other facets of autistic symptomatology. However, the specifics of how syllable-evoked P1m interplays with the severity of autistic symptoms remain uninvestigated.

3. Since one of the implications of this study is to use MEG neurobiomarkers for early identification, it’s important to include information about when ASD symptoms are presented and the age at which children were usually diagnosed with ASD.

Thank you for emphasizing the significance of highlighting the onset of ASD symptoms and the typical age at diagnosis. Your comment is crucial in contextualizing the importance of early identification, especially considering the potential application of MEG neurobiomarkers in this endeavor. In response to your invaluable feedback, we highlighted the following:

Incorporation of Age and Symptom Onset: In the introduction, we have expanded upon the early presentation of ASD symptoms, elucidating the challenges often encountered during the initial stages of diagnosis.

Highlighting Diagnostic Delays: We have provided specific details from recent surveillance studies that underline the difference between the initial onset of caregiver concerns and the comprehensive evaluation and subsequent official diagnosis.

Factors Affecting Diagnosis: To give readers a comprehensive understanding, we have also elaborated on the myriad factors that can potentially delay or complicate the diagnostic process, including the subtlety of developmental milestones, coexisting conditions, and socioeconomic and cultural barriers.

Importance of Early Intervention: We have emphasized the critical benefits of timely interventions like the Early Start Denver Model, emphasizing how they substantially improve outcomes when initiated early.

We sincerely hope these changes effectively address your concern, providing readers with a deeper understanding of the diagnostic landscape and underscoring the potential contributions of our study. We are grateful for your feedback, which has been instrumental in enhancing the quality and clarity of our manuscript.

(Introduction)

Autism spectrum disorder (ASD) is a neurodevelopmental disorder characterized by impaired social interaction and communication along with restricted and repetitive behavioral patterns and fixated interests, as defined in the Diagnostic and Statistical Manual of Mental Disorders, Fifth Edition (DSM-5) [1]. Early diagnosis and intervention are vital for optimizing outcomes in individuals with ASD [2-4]; However, clinical diagnosis of ASD in young children can be challenging, as the characteristic symptoms may be less evident during the early developmental stages. Surveys of families with children affected by ASD highlight common delays between the initial emergence of caregiver concerns and the comprehensive evaluation as well as between the evaluation and official ASD diagnosis [5-7]. Notably, a recent multicenter surveillance study reported that, while 85% of caregivers noted concerns regarding developmental delays by 36 months of age, only 61% of the children underwent a comprehensive evaluation by 48 months. The median age at diagnosis was 52 months [3]. 

Diagnosing ASD proves challenging due to several factors including time constraints during office visits, the subtle nature of social developmental milestones, and the variability of signs and symptoms observed in individual children. The process can be further complicated by numerous elements that potentially delay the diagnosis, including the presence of less severe symptoms, female gender, concurrent issues such as anxiety or hyperactivity, lack of continuous care, and others such as socioeconomic factors and language barriers [7-10]. Moreover, the symptoms can be obscured or exacerbated by coexisting problems, which may affect both the timing and accuracy of the diagnosis. The lapse in establishing a timely diagnosis is clinically concerning as it might postpone the implementation of evidence-based behavioral interventions, potentially leading to suboptimal outcomes [11]. Implementing interventions such as the Early Start Denver Model, a behavioral therapy specifically designed for children with ASD, has been shown to enhance social, language, and cognitive functions, especially when initiated before the age of 5 (between 12 and 60 months) [12-14]. These findings underscore the importance of early diagnosis and intervention to improve the prognosis and quality of life of individuals with ASD. Given the fluctuating and sometimes elusive nature of behavioral autistic traits highlighted above, delving into the biological and physiological characteristics of ASD may forge a path towards more precise diagnostics and nuanced evaluations of treatment responses.

Methods and results:

1. Did the authors conduct any sample size estimation based on their previous study?

Thank you for your insightful inquiry regarding sample size estimation based on our previous studies.

In response to your comment, in the Methods section, we have elaborated on our approach to determining the sample size. The rationale behind our sample size estimation is rooted in a preliminary analysis from a subset of data taken from our prior research. To ensure robustness in our investigation, we utilized a preliminary sample from our past studies to estimate the effect size using a squared multiple correlation coefficient (R²). This estimation produced specific R² values based on the P1m intensities from either hemisphere. To ensure a conservative approach, we proceeded with the smaller of the R² values. This decision was pivotal in determining our required sample size. Our chosen effect size, alpha level, and power ensured that we had an adequate sample to detect potentially significant effects, even considering potential data exclusions or unforeseen challenges.

We employed G*Power, a widely recognized tool, for our sample size computations to ensure precision and reliability. We hope this detailed account clarifies our procedure and the rationale behind it, emphasizing its grounding in prior data and established statistical methodologies. We truly appreciate your attention to detail, and we believe this added explanation will significantly enhance the rigor and clarity of our research methodology for readers.

(Methods)

In this study, we evaluated the severity of autism symptoms in child participants, both with and without ASD, using the SRS [63]. We assessed their intelligence using the Japanese version of the K-ABC [59]. The syllable-evoked P1m data were derived from MEG recordings. Our primary goal was to investigate the relationship between autistic symptoms, as indicated by the total T-scores on the SRS, and the intensity of the P1m response across children with and without ASD. To capture a comprehensive view of this relationship, we employed a multiple linear regression model. This model considered the possible influence of fluid intelligence, as measured by the Mental Processing Scale (MPS) from the K-ABC, on autism symptoms [64].

Specifically, our model aimed to predict the total T-scores of the SRS based on the (log-transformed) intensity of P1m from either the left or right hemisphere and the MPS scores in the K-ABC. To determine the required sample size for this investigation, we began by estimating the effect size using a squared multiple correlation coefficient (R²) based on a preliminary sample [65]. This sample comprised data from six TD children from our prior studies [29, 56]. Our preliminary analysis, conducted on this sample, produced R² values of 0.365 and 0.469 for models considering the right and left P1m (log-transformed) intensities, respectively. To be conservative, we chose to proceed with the smaller R² value of 0.365. Setting the alpha at 0.05 and the power (1 - beta) at 0.80, we arrived at an effect size F² of 0.574 [66], determining a total sample size of 21 to accommodate the two predictors. We used G*Power version 3.121.6 [67, 68] for this sample size computation. We concluded to enlist at least 25 participants in each group. This margin would accommodate potential exclusions for any unforeseen reasons.

2. The first sentence in the result section seems to be cut out.

Upon reviewing the manuscript, we identified the inadvertent truncation of the first sentence during our revision process. We deleted this sentence from the manuscript. We sincerely apologize for any confusion this may have caused and appreciate your diligence in ensuring the coherence and clarity of our manuscript.

Discussion:

1. Please provide more rationales for the clinical implication. For example, how the findings could be used as an early biomarker, and how this neurobiomarker could be used to support early intervention.

In response to your feedback, and acknowledging the limitations identified in your comments, we revised the manuscript so that it does not propose a definitive assertion regarding the immediate use of our findings as a direct biomarker for ASD. Instead, in discussion of the revised manuscript, we have focused on interpreting our observations in the context of existing research.

Our study is exploratory in nature, and, while we are excited about its potential implications, we also approach them with caution. We believe that only after rigorous validation across multiple, larger studies can these findings be directly translated into clinical practice. Thank you for pushing us to think deeply about the broader relevance of our work. Your insights are invaluable in ensuring our research is both rigorous and meaningful. 

Reviewer #2: Hirosawa and colleagues investigated in children with ASD (~3 to 7 years) and control typically developing (TD) children auditory responses to speech stimuli (syllable /ne/) using MEG. Based on their previous results the authors expected to find in children with ASD atypical amplitudes of the transient component P100 in the left and right hemispheres (increased in the right hemisphere, decreased in the left), as well as correlations of these amplitudes with autism severity estimated using SRS questionnaire.

Stimuli were presented using passive oddball paradigm. Deviant /ne/ (n=90) differed from standard /ne/ (n=456) in intonation. Duration of each stimulus was 342 ms, including 65 ms of consonant. ISI was 818 ms. The ‘zero time’ was set at the beginning of the vowel /e/. -50 to 0 ms was used as a baseline (i.e. overlapping with /n/ presentation).

Data were recorded using child-size MEG system. No individual brain models were available. Standards were used to fit the dipole model to P100, but only P100 to deviants was analyzed.

The main result is a negative correlation between P100m amplitude in the right hemisphere and SRS score in ASD.

The positive side of this study is the use of the child-size system, which allows better sensitivity to auditory magnetic fields in children. The relatively large sample of ASD participants (N=49) is also an advantage. However, I have a number of concerns.

Major issues.

1. In the Introduction and methods the authors claim that P100 may serve as a biomarker of ASD. However, abnormalities in auditory responses may be not specific to ASD, but associated with ADHD or auditory processing disorder, since both these disorders have high comorbidity with ASD.

Thank you for your observation regarding the potential non-specificity of auditory-evoked potentials, particularly the P1m (or P50) suppression, to ASD. We completely concur that certain abnormalities in auditory responses may also be associated with conditions such as ADHD or auditory processing disorder, which often co-occur with ASD. This potential overlap underscores the necessity of rigorous differentiation when proposing neural markers for specific disorders. In response to your valuable feedback, we have introduced a comprehensive paragraph in the manuscript clarifying the nature of the P1m response; its various nomenclatures; its developmental patterns; and most importantly, its significance in disorders other than ASD. This addition emphasizes that, while P1m suppression has been linked to disorders like schizophrenia and ADHD, our primary focus in this study lies in evaluating the latency and intensity of the P1m response and its potential relation to ASD. We hope that this expanded context adequately addresses your concerns.

(Introduction)

Auditory-evoked potentials (AEPs in EEG recordings and AEFs in MEG recordings) represent the auditory system's electromagnetic signals, generated in response to sound stimuli. These signals, precisely captured through EEG or MEG, are distinguished based on their occurrence time post-stimulus onset: early responses within 10 ms, middle-latency responses between 10 and 50 ms, and long-latency responses between 60 and 500 ms [20]. The long-latency response, especially notable in the cortical region, can be outlined using averaging techniques to enhance the target response's signal-to-noise ratio (SNR) [21-23]. 

Central to our discussion is the long-latency response of AEF, characterized by three notable peaks at approximately 50 ms (P1m), 100 ms (N1m), and 200 ms (P2m). Within these, the first and second peaks are often the focus of examination. However, it is important to note that, in children younger than 10 years old, the second peak may not be fully developed and might be less discernible, making the first peak a more reliable measure to assess the auditory cortex response in this age group [24-27]. In early childhood, the latency of these peaks deviates significantly from adult patterns, resulting in varied nomenclatures for the first peak across different studies, including P1m [28, 29], M50 [25, 30, 31], P50m [32], and P100m [33]. In this study, we will adopt the term P1m for consistency. The P1m, primarily generated by neural activity in the primary and associative auditory cortices [34], acts as a pre-attentional response, reflecting the developmental status of the central auditory pathways. Specifically, its amplitude and latency indicate neural synchrony and auditory stimulation transmission time, respectively [35-37]. 

A distinctive feature of the P1m response is its "suppression" phenomenon. This suppression is often termed "P50 suppression" despite referring to the same neural response as P1m. In this context, P50 suppression is a type of sensory gating observed under specific conditions, signifying a diminished amplitude of the response to a second stimulus (S2) relative to a first (S1). Impairments in this suppression are frequently linked to sensory gating deficits observed in disorders like schizophrenia and ADHD [35, 38-42]. However, the feasibility of P50 (or P1m in our terminology) suppression as a consistent biomarker for ASD is yet to be firmly established. While some studies hint at altered latency and intensity of P50 responses in individuals with ASD (as described below), the majority have not identified significant P50 suppression deficits, particularly in high-functioning children and adults with autism [43-46]. Despite one study noting impaired P50 suppression in young children with autism, later research, including larger sample studies, has not affirmed these findings [45-47]. Given this backdrop, a detailed investigation into this neural response in ASD, particularly assessing potential changes in latency and intensity, stands out as a crucial direction for future research.

2. The authors mentioned that part of their sample overlapped with the sample included in their other studies (e.g. Yoshimura, Mol Autism. 2013;4(1):38.), where atypical lateralization of P100 (called P50 in that study) was found. However, the authors did not specify the number of overlapping participants in each sample. Will the results be reproduced when only ‘new’ subjects are analyzed?

Thank you for bringing to light the essential aspect of sample overlap between our current study and our previous work, specifically the one cited (Yoshimura, Mol Autism. 2013;4(1):38.). We understand the potential implications of such overlaps for the interpretability and generalizability of our results. In response to your comment, we have clearly indicated in the revised manuscript the number of newly recruited participants. We have also reanalyzed our data exclusively with the newly recruited participants, as you rightly suggested. While we decided not to include these specific results in the main manuscript to avoid redundancy, we have included them separately for your review (S6 Tables for Reviewer 2). This will allow you to compare the outcomes from our full sample with the results derived solely from the new participants. Your suggestion to re-evaluate our data with only the new participants ensures our analysis is thorough and aids in gauging the robustness of our findings. We are hopeful that these adjustments will address your concerns and further fortify the validity of our study. Thank you for your meticulous feedback, which undoubtedly enhances the quality and clarity of our work. 

(Methods)

 We recruited participants from Kanazawa University and affiliated hospitals, securing 57 children with ASD and 26 TD children. The diagnosis of ASD followed the criteria in the Diagnostic and Statistical Manual of Mental Disorders, Fourth Edition (DSM-IV) [15], utilizing either the Diagnostic Interview for Social and Communication Disorders (DISCO) [16] or the Autism Diagnostic Observation Schedule (ADOS) [17]. To mitigate the potential confounding effects of intellectual disability, we excluded six children with ASD who scored below 70 on the K-ABC Mental Processing Scale. Additionally, two children with ASD were excluded due to missing data concerning head location during the MEG recording. Consequently, our study included 49 children with ASD (37 boys and 12 girls; aged 40-92 months) and 26 TD children (21 boys and 5 girls; aged 42-89 months). Table 1 presents the characteristics of the participants. 

The Research Center for Child Mental Development at Kanazawa University (https://kodomokokoro.w3.kanazawa-u.ac.jp/en/) continuously recruits children both with ASD and TD children as part of the research initiative known as the "Bambi Plan," which focuses on ASD research. Our participant pool was drawn from individuals recruited at this center between the years 2009 and 2014. We accessed their data between September 1 and September 30, 2022, for research purpose and had access to information that could identify individual participants during or after data collection. Notably, there was an overlap in the participant pool with some of our previous studies [18-22]. We integrated all available data from these earlier studies, which included 8 TD children (7 boys and 1 girl, aged 42–75 months) and 21 children with ASD (19 boys and 2 girls, aged 40–92 months), supplementing it with new participants. While there was an overlap in the data, the focal points and results of the current study are distinct from those of previous research. Exclusion criteria were defined, ruling out potential participants with (1) blindness, (2) deafness, (3) any other neuropsychiatric disorder, or (4) an ongoing medication regimen. Written informed consent was obtained from parents of the children prior to their participation in the study. The Ethics Committee of Kanazawa University Hospital approved the methods and procedures, all of which were conducted in accordance with the Declaration of Helsinki.

3. The argumentation used in the manuscript is fully based on the correlational approach, without any reflection about putative neural mechanisms underlying ASD vs TD differenses in P100, its lateralization and correlation with SRS. I doubt that this correlational approach can supply reliable biomarker (also see point 1).

In response to your feedback, and acknowledging the limitations identified in your comments, we revised the manuscript so that it does not propose a definitive assertion regarding the immediate use of our findings as a direct biomarker for ASD. Instead, in the discussion of the revised manuscript, we have focused on interpreting our observations in the context of existing research.

Our study is exploratory in nature, and, while we are excited about its potential implications, we also approach them with caution. We believe that only after rigorous validation across multiple, larger studies can these findings be directly translated into clinical practice. Thank you for pushing us to think deeply about the broader relevance of our work. Your insights are invaluable in ensuring our research is both rigorous and meaningful.

4. It is unclear why standards were used to fit the dipole, but only responses to deviants were analyzed. If the authors expect differences in P100 only for deviants, they need to explain why.

Thank you for pointing out the need for clarity in our methodological approach concerning the selection of stimuli for dipole fitting and analysis. In our study, we utilized only the responses to the standard stimuli for both the dipole fitting and subsequent analysis. We understand that our original presentation might have led to ambiguity, and we appreciate your attention to detail in pointing this out. To address this, we have added sentences to the Results section that explicitly state, "Only responses to the standard stimuli yielded a sufficient number of epochs for accurate ECD calculation across participants. Therefore, only responses to standard stimuli were analyzed."

We trust this addition offers clarity and ensures that the methodology is unmistakably presented. The decision to use only responses to standards was based on achieving a reliable number of epochs for accurate analysis across participants. We appreciate your feedback, which aids in ensuring our study is both comprehensive and transparent.

(Results)

MEG data were collected from 26 TD children and 49 children with ASD. Only responses to the standard stimuli yielded a sufficient number of epochs for accurate ECD calculation across participants. Therefore, only responses to standard stimuli were analyzed. Eleven participants did not achieve the minimum of 300 epochs. Specifically, data from two TD children and nine children with ASD were excluded. The numbers of participants excluded were not significantly different between groups (χ2 = 1.55, p = 0.214). Thus, the responses were averaged over 403.8 ± 2.2 (mean ± SD) epochs for the 24 TD children and 382.4 ± 39.1 epochs for the 40 children with ASD. A Student’s t-test revealed that the average number of epochs was significantly higher for the TD children (t(62) = 2.26, p = 0.03). 

5. Number of averages per subject is not specified. Is this number differs in ASD and control groups? Is the correlation still significant when SNR is taken into account, e.g. by correcting for square root of number of averages?

Thank you for raising the important point regarding the specification of averages per participant and accounting for signal-to-noise ratio (SNR). Your insights were valuable in refining our manuscript.

Number of Averages Per Participant: We have now specified the number of averages per participant in the Results section. The responses were averaged over 403.8 ± 32.2 epochs for the 24 TD children and 382.4 ± 39.1 epochs for the 40 children with ASD. A subsequent t-test indicated a significantly higher average number of epochs for the TD children.

Incorporating SNR: Given the observed difference in the number of epochs between the ASD and TD groups, and in recognition of the potential influence of SNR on our results, we reanalyzed our data with the SNR factored in. Specifically, we used the square root of the number of averages as a proxy for SNR and integrated it into our regression models, according to your suggestion. Notably, even after accounting for SNR, the left P1m latency persisted as a significant predictor, both generally and within the TD group. This adjustment ensures the robustness of our findings and that the relationships observed are not merely artifacts of variations in SNR.

These revisions aim to address your concerns comprehensively. We appreciate your suggestions, which led to the enhancement of our analysis and the clarity of our presentation.

(Results)

 MEG data were collected from 26 TD children and 49 children with ASD. Only responses to the standard stimuli yielded a sufficient number of epochs for accurate ECD calculation across participants. Therefore, only responses to standard stimuli were analyzed. Eleven participants did not achieve the minimum of 300 epochs. Specifically, data from two TD children and nine children with ASD were excluded. The numbers of participants excluded were not significantly different between groups (χ2 = 1.55, p = 0.214). Thus, the responses were averaged over 403.8 ± 2.2 (mean ± SD) epochs for the 24 TD children and 382.4 ± 39.1 epochs for the 40 children with ASD. A Student’s t-test revealed that the average number of epochs was significantly higher for the TD children (t(62) = 2.26, p = 0.03). 

(Results, Significant relationship between left P1m latency and SRS T-scores)

Given the significant difference in the average number of epochs between the ASD and TD groups, we introduced another regression model to account for the SNR. We used the square root of the number of averages for each participant as a proxy for SNR [86] and added it to the original regression model. In particular, this revised model predicts the SRS total T-score based on the left P1m latency, diagnostic group (ASD or TD), MPS scores, interaction between left P1m latency and diagnosis, and square root of the number of averages. By integrating the SNR proxy, we ensure relationships observed with left P1m latency are not only a result of SNR variations. Even with this adjustment, the left P1m latency remained a significant predictor (t(53) = −2.62, p = 0.011). Adopting a similar approach for the separate group-based regressions, the left P1m latency remained a significant predictor (t(17) = −2.61, p = 0.018) within the TD group. All other factors remained nonsignificant. Detailed results are provided in S3 Table. 

(Results, More pronounced autistic symptoms are associated with stronger leftward lateralization in P1m intensity, exclusively in children with ASD)

Using the approach previously described and recognizing the significant difference in epoch averages between the ASD and TD groups, we adjusted our model to factor in the SNR, using the square root of averages as an SNR proxy [87]. Post-adjustment, the interaction between leftward lateralization and diagnosis retained its significance (t(45) = −2.38, p = 0.022). Similarly, in the adjusted regression for the ASD group, the effect of leftward lateralization remained significant (t(26) = 2.81, p = 0.009). Comprehensive results of this modified analysis are available in S4 Table.

6. Please, present more visual information. This would be of great help for the reader to understand/interpret the results. I would be interested to see:

(6-)1. waveforms of the standard and deviant stimuli,

(6-)2. timecourses of the responses to standards and deviants (including essentially long baseline period before /n/),

(6-)3. violin plots of P100 latensies in both groups,

(6-)4. violin plots of P100 amplitudes in both groups

Thank you for your constructive feedback emphasizing the inclusion of visual information. We agree that visual representation can significantly aid in understanding and interpreting the results.

To address your recommendations:

(6-1) Waveforms of the Standard and Deviant Stimuli: We have included Figure 1, which shows the waveform of the standard /ne/ and the deviant /Ne/ voice stimulus. This figure clearly demonstrates the segmented portions of each stimulus, providing clarity on the durations and the defining onset time for MEG analysis.

(6-2) Timecourses of the Responses: Figure 2 has been incorporated to visually represent the neuromagnetic response to the standard syllable /ne/ stimuli. Responses to the deviant stimuli are not shown because the number of epochs is not sufficient and it is inappropriate to add them up and average them. 

(6-3 & 6-4) Violin Plots for P1m Latencies and Amplitudes: To give a detailed visual insight into the distribution of P1m latency and intensity values, Figure 3 has been introduced. This figure comprises violin plots for both the right and left hemispheres, separated by diagnostic groups (ASD and TD). This visual representation offers a clear comparative perspective on P1m latencies and intensities across groups and hemispheres.

We believe that these additions significantly enhance the manuscript's clarity and comprehensibility. We are grateful for your feedback that led us to present our findings in a more digestible and visually appealing manner. We hope these visual aids address your concerns and improve the reader's experience.

Fig 1. Waveform of the standard /ne/ (left) and deviant /Ne/ (right) voice stimulus. 

The total duration of each stimulus was 342 ms, segmented into 65 ms for the consonant /n/ and 277 ms for the subsequent vowel sound /e/. The MEG analysis onset time was defined as the beginning of the vowel portion. It is important to note that only the standard stimuli were used for the subsequent equivalent current dipole (ECD) estimation, as only this condition provided a sufficient number of epochs for accurate ECD calculation.

Fig 2. Neuromagnetic response to the standard syllable /ne/ stimuli.

Syllable-induced AEF with a baseline from −50 to 0 ms relative to the onset of the vowel /e/. The resultant AEF displayed a pronounced activity peak between 45 and 150 ms. The onset of the consonant /n/ is at −65 ms relative to that of /e/. The blue arrow displays the direction of the estimated dipole moment.

Fig 3. Violin plots of P1m latency and log-transformed intensity by diagnostic group.

The figure displays violin plots illustrating the distribution of P1m latency and log-transformed intensity values for both the right (R) and left (L) hemispheres, separated by diagnostic group (ASD and TD). The top two plots represent the right hemisphere with the first showcasing P1m latency and the second depicting the log-transformed P1m intensity. The bottom two plots correspond to the left hemisphere; the first illustrates P1m latency, and the second demonstrates the log-transformed P1m intensity. The width of each "violin" indicates the density of the data at different values, offering a visual representation of the data's distribution.

7. I wonder why interval -50 to 0 ms relative to the onset of /e/ vowel was chosen as the baseline? There is a risk that this baseline interval itself differs in ASD and control group. Will the result still keep if the baseline is taken before the /n/ consonant?

Thank you for pointing out the discrepancy concerning the setting of our baseline in relation to the onset of the /e/ vowel. Fortunately, a reviewer's question led to the discovery that we had made a mistake in the analysis procedure. In the first manuscript, the baseline was mistakenly set to −50 ms to 0 ms relative to the onset of the /n/ consonant, which was an inadvertent error. We genuinely apologize for this mistake and any confusion it may have caused.

To correct this error, we revisited our data and recalculated the P1m parameters by setting the correct baseline relative to the onset of the /e/ vowel. Unfortunately, the results do not mirror those in the original manuscript. Reflecting these results, we have rewritten the Methods, Results, and Discussion sections to reflect the accurate findings obtained from the correct baseline setting. The revised manuscript paper has baseline set to −50 ms to 0 ms relative to the onset of the vowel /e/ as we originally intended. The reason for baseline correction for the vowel /e/ in this study was to maintain consistency with our previous studies. Until now, in our past research, we have considered it a response to the vowel /e/ based on the peak latency of the P1m. However, as the reviewer rightly points out, there may also be responses to the consonant /n/, and the proportion of brain response to the /n/ in both TD and ASD may potentially affect the results in this study. To confirm this, it would be necessary to study both /ne/ and /e/ stimuli. Unfortunately, this study only includes /ne/ stimuli. Therefore, we will acknowledge this limitation in the Discussion section. 

Despite the baseline correction, the insights derived from this recalibrated data remain critical. While the outcomes have shifted, they continue to provide valuable insights but also highlight the intricate nature of neural mechanisms and their relationship with autistic traits. The core findings, even after adjustments, offer substantial contributions to our understanding of the topic.

(Methods, Equivalent current dipoles for the AEFs)

 To calculate the ECDs without using MRI-based anatomical information, we adapted a spherical model to represent the volume conductor, positioning it at the center of the MEG helmet. The onset time of the syllable stimuli (designated as 0 ms) was set at the onset of the vowel /e/ rather than the consonant /n/ in accordance with our prior studies [29, 56-58, 62]. We then averaged the time series spanning from -150 ms to 1000 ms (with a minimum of 300 epochs for standard stimuli) for every sensor, after baseline correction. The baseline was set from −50 ms to 0 ms relative to the onset of the vowel /e/. Artifact-contaminated segments (such as eye blinks, eye movements, and bodily motions, typically exceeding ± 4 pT) were omitted from the analysis. A uniform ECD model facilitated the computation of the current sources, engaging at least 42 sensors per hemisphere [80]. All procedural steps and parameters were aligned with our earlier studies [29, 56-58, 61, 62], ensuring the findings from this research can be directly compared with our prior results. However, it is worth noting that our chosen baseline, spanning from −50 ms to 0 ms relative to the onset of the vowel /e/, might introduce variability in the baseline interval between the ASD and control groups. The results could differ if using an alternative baseline, possibly preceding the consonant /n/.

(Results)

Fig 2. Neuromagnetic response to the standard syllable /ne/ stimuli.

Syllable-induced AEF with a baseline from −50 to 0 ms relative to the onset of the vowel /e/. The resultant AEF displayed a pronounced activity peak between 45 and 150 ms. The onset of the consonant /n/ is at −65 ms relative to that of /e/. The blue arrow displays the direction of the estimated dipole moment.

(Discussion)

Another limitation arises from setting the baseline relative to the onset of the vowel /e/. Brain responses to the preceding consonant /n/ may exist, and the proportion of participants demonstrating this response might differ between the TD and ASD groups, potentially influencing the results. To validate this, future research should include both /ne/ and /e/ stimuli. 

We assure you that all necessary precautions will be taken in future research endeavors to prevent any such oversights. We are grateful to you for highlighting this issue, as it allowed us to correct and refine our study. Your feedback has been invaluable in enhancing the rigor and accuracy of our research. 

8. Was the head position continuously tracked during the experiment and corrected to a common position? Was the predominant head position different in ASD and control groups? For example, could differences in P100 amplitude (increased on the right and decreased in the left in ASD) be explained by differences in the head position between the groups?

We appreciate the reviewer's helpful and important comments. The positions of the marker coils were obtained at least once before and/or after the MEG recording. Periods with obvious body movement have been excluded based on noise detection, and, in cases where there was a significant shift in head position during the session, they were excluded due to a decrease in the goodness of fit (GOF) in the dipole analysis. Whether the head's position itself has an impact was not investigated in this study. We will mention this as a limitation.

(Discussion)

In this study, participants were monitored using a video camera to detect noticeable body movements. An examiner accompanied the children in the shielded room and instructed them to maintain a constant head position throughout the experiment. Instances of pronounced body movement were excluded based on noise detection. Additionally, participants who exhibited significant shifts in head position during the session were excluded due to a reduction in the GOF in the P1m dipole analysis. However, this study did not account for the impact of fine head movements and variations in head shape.

9. Statistical analysis. It is unclear why the authors predict SRS with P100 amplitude, MRS and diagnosis. The meaning of significant interaction between P100 and Diagnosis is unclear.

Why not to perform ANOVA with factors Group and Hemisphere, as the authors did in their previous study (Yoshimura, 2013 MolA)?

Our analytical approach in this study was predominantly driven by our formulated hypotheses. As detailed in the revised manuscript's introduction, we postulated a correlation between reduced intensity of the syllable-evoked P1m, especially in the left hemisphere, and heightened autistic traits. Furthermore, prompted by your suggestion and upon reanalyzing all data after obtaining P1m parameters from the correct baseline, we also formulated the hypothesis that a decreased leftward lateralization in the intensity of this response signifies more pronounced autistic traits.

The linear regression model, as opposed to the ANOVA model we utilized in our prior study (Yoshimura, 2013 MolA), was deemed more fitting due to its ability to examine these specific relationships. The interaction term between P1m intensity and diagnosis in the model is important as it aims to capture potential variations in the correlation between P1m intensity in both hemispheres and autistic traits between the two groups. The rationale behind this model, as well as the significance and meaning of the interaction term, is now delineated in greater detail in the Statistical analysis section of the revised manuscript.

Regarding the ANOVA model as employed in our prior study, while the ANOVA approach offers its own merits, we believed that the linear regression model better addressed the current study's objectives. However, we appreciate your suggestion and understand the value of consistency across studies.

(Introduction)

 Here, we explicitly acknowledge the exploratory nature of the present study. Accordingly, our hypotheses are formulated on provisional grounds: (i) Stronger intensity of the syllable-evoked P1m in the left hemisphere corresponds with better conceptual inference skills among TD children [11]; (ii) diminished conceptual inference skills potentially reflect certain facets of autistic symptomatology [62]; and (iii) TD children typically display a leftward lateralization in syllable-induced P1m, which is characterized by a more pronounced intensity in the left hemisphere compared to the right. Additionally, this lateralization seems to be subdued in children with ASD [29, 56]. Given these observations, we postulate that a reduced intensity of syllable-evoked P1m, especially in the left hemisphere, correlates with more pronounced autistic traits. Furthermore, a decreased leftward lateralization in the intensity of this response—potentially indicated by a diminished intensity in the left hemisphere coupled with an augmented intensity in the right—also signifies more accentuated autistic traits. To validate our hypothesis, we propose employing linear regression models to predict the degree of autistic traits, as denoted by scores on the SRS [63], using the data derived from the intensity measurements of the syllable-evoked P1m in both hemispheres. Additionally, we aim to assess any correlation between P1m latency and the severity of autistic traits. The knowledge gleaned from this investigation holds promise for substantially influencing the clinical approach towards diagnosing and managing ASD. By identifying a more objective and noninvasive metric for autistic traits, our ambition is to set the stage for more timely diagnosis and early interventions.

(Statistical analysis)

Our primary hypothesis was that a reduced intensity of syllable-evoked P1m, especially in the left hemisphere, correlates with more pronounced autistic traits. Additionally, a decreased leftward lateralization in the intensity of this response also signifies more pronounced autistic traits. 

First, to evaluate this hypothesis, our statistical model aimed to explore any potential differential correlation between the intensity of syllable-evoked P1m and autistic traits between children with ASD and TD children. Recognizing the potential influence of intellectual abilities on SRS scores, as suggested in prior research [81], we incorporated this variable into our model. Thus, we performed a linear regression analysis predicting the SRS total T-score based on the intensity of the left (or right) P1m, diagnostic category (ASD or TD), MPS scores, and an interaction term between P1m intensity and diagnosis. This interaction term is vital for discerning potential variations in the relationship between P1m intensity in the respective hemispheres and autistic traits among the two groups. The inclusion of MPS scores mitigates the potential confounding effect of intelligence on SRS scores. We conducted this regression twice—once for the left hemisphere and once for the right—without adjusting for multiple comparisons, given the few preplanned and likely correlated comparisons [82, 83]. Consequently, we set the statistical significance level at p < .05. Aligning with our previous methods [29, 56], we used log-transformed intensity values rather than raw intensity for comparisons. For thoroughness, we also executed the same analysis using P1m latency in place of log-intensity.

Second, to delve deeper into the relationship between syllable-induced P1m intensity and autistic traits, we sought to discern how changes in the P1m intensity, either diminished in the left hemisphere or amplified in the right, might correlate with more pronounced autistic traits within each diagnostic group (i.e., TD children and children with ASD). We performed separate linear regression analyses for each group predicting the SRS total T-score based on P1m intensity (either left or right) and MPS scores. This analysis was undertaken four times—once for each hemisphere within both diagnostic categories—without corrections for multiple comparisons, setting the significance threshold at p < .05, given the preplanned and potentially correlated comparisons [82, 83]. For completeness, P1m latency was also examined in lieu of log-intensity.

Third, we sought to identify any varying correlations between the degree of leftward lateralization in P1m intensity and autistic traits across children with ASD and TD children. A linear regression analysis was undertaken to predict the SRS total T-score based on the P1m's leftward lateralization (defined as log-transformed P1m intensity in the left hemisphere minus its counterpart in the right), diagnostic category, MPS scores, and an interaction between P1m leftward lateralization and diagnosis. Similarly, we conducted an analysis replacing log-intensity with P1m latency with a significance level set at p < .05, following the rationale of limited preplanned and potentially interlinked comparisons [82, 83].

Fourth, we aimed to probe the association between the leftward lateralization of syllable-induced P1m and autistic traits within each diagnostic group. We performed separate linear regression analyses for each group predicting the SRS total T-score based on P1m's leftward lateralization and MPS scores. This analysis phase was executed twice, once for each diagnostic category, again without multiple comparison corrections. The statistical significance threshold was retained at p < .05 due to the limited preplanned and potentially correlated comparisons [82, 83]. As a thoroughness measure, we also examined P1m latency as an alternative to log-intensity.

10. lines. 319-322 ‘We further analyzed the association between right P1m intensity and the SRS-subscales and found a significant negative correlation with the SRS autistic mannerism T-score, suggesting that the relationship between P1m intensity and ASD symptoms may be primarily driven by autistic mannerisms.’

To make this conclusion the authors need to insure that correlation with mannerism subscale is significantly different from correlations with other subscales.

Thank you for highlighting this point. Upon reanalyzing our data due to corrections in the baseline setting, as mentioned in response to prior comments, the results pertaining to the correlation between right P1m intensity and the SRS subscales have changed. Consequently, the section you have referenced has been removed from the revised manuscript. We appreciate your thorough review and the diligence with which you have pointed out areas of potential concern. This has greatly assisted us in improving the rigor and clarity of our findings.

Minor issues

Fig.1. ‘Margins are statistics that are calculated based on a dataset where some or all of the covariates are held at fixed values different from their true values [49, 50).’

The Legend is not very helpful, as the reader needs to address citations 49, 50 to understand the plot. Please explain meaning of these margins more clearly.

Considering violation of homogeneity of variance, another approach would be to calculate partial non-parametric correlations, controlling for MRS/IQ.

Thank you for pointing out the ambiguity in the legend of Figure 1. We agree that the initial explanation may have been too concise and potentially unclear. In the revised manuscript, we have aimed to clarify the interpretation and meaning of the margins in the legends for Figures 4 and 5 without requiring the reader to refer to the cited papers directly. We value your feedback and believe that these modifications have improved the clarity and rigor of our manuscript.

Fig 4. Adjusted predictions of SRS total T-score over a range of latency of P1m in the left hemisphere.

This figure illustrates the predicted SRS total T-scores as a function of the latency of P1m in the left hemisphere. These predictions stem from a regression model, applied within TD children, that predicts SRS total T-scores based on the latency of P1m in the left hemisphere and MPS scores. The analysis employed robust standard errors to account for potential heteroscedasticity.

The horizontal axis showcases the latency of P1m in the left hemisphere, ranging from 40 to 120 ms. The vertical axis corresponds to the model's predicted SRS total scores. The plotted line depicts the variation in predicted SRS scores as a function of the P1m latency in the left hemisphere. These predictions were made across different latencies of P1m while keeping the MPS scores constant at the average for TD children (i.e., 100.87). Confidence intervals around predictions were informed by standard errors, which were determined using the delta method [85, 86].

SRS, social responsiveness scale; TD children, typically developing children; MPS, mental processing scale

Fig 5. Adjusted predictions of SRS total T-score by diagnostic group over a range of leftward lateralization of log-intensity.

This figure visualizes the predicted SRS total T-scores for ASD and TD children, derived from a regression model accounting for P1m's leftward lateralization (defined as the log-transformed P1m intensity in the left hemisphere minus that in the right), diagnostic group, MPS scores, and the interaction between P1m leftward lateralization and diagnosis. Robust standard errors were used in this analysis to account for potential heteroscedasticity.

The horizontal axis spans a range of leftward lateralization from −1.00 to 1.00. The vertical axis represents the model's predicted SRS total scores. Each line corresponds to a diagnostic group, showing how predicted SRS scores vary across P1m’s leftward lateralization. Predictions were made at given value of P1m's leftward lateralization while holding MPS scores constant at the sample mean (i.e., 102.38). Standard errors, derived using the delta method [85, 86], inform the plotted confidence intervals around predictions.

SRS, social responsiveness scale; ASD, autism spectrum disorder; TD children, typically developing children; MPS, mental processing scale

Please report high-pass filter, even if only an inbuilt filter was applied and no additional filtration was used.

Thank you for your comment regarding the high-pass filter specification. In the revised manuscript, we have now provided the necessary information regarding the inbuilt filter used during the MEG recordings. Specifically, we collected bandpass-filtered MEG data in the range of 0.16–200 Hz at a sampling rate of 1000 Hz, as detailed in the MEG recordings section under Methods. We appreciate your attention to detail and ensuring our methods are clearly and comprehensively described.

(Method, MEG recordings)

MEG recordings were conducted for 12 minutes during the presentation of stimuli, and bandpass-filtered MEG data (0.16–200 Hz) were collected at a sampling rate of 1000 Hz.

Line 260: Sentence starts with small ‘i’.

Upon reviewing the manuscript, we identified the inadvertent truncation of the first sentence during our revision process. We delete this sentence from the manuscript. We sincerely apologize for any confusion this may have caused and appreciate your diligence in ensuring the coherence and clarity of our manuscript.

Abbreviation MRS is mot explained in the text.

Thank you for pointing out the oversight regarding the abbreviation "MPS." We have now amended the manuscript to introduce and spell out the "Mental Processing Scale (MPS)" at its first appearance in the Experimental design and sample size calculation section. We appreciate your feedback ensuring clarity and comprehensiveness throughout the manuscript.

(Experimental design and sample size calculation)

This model considered the possible influence of fluid intelligence, as measured by the Mental Processing Scale (MPS) from the K-ABC, on autism symptoms [64].

Line 201-202: ‘Participants received the stimulus through both ears via a gap in the MEG chamber, which was transmitted by loudspeakers…’

This is unclear, please explain.’

Thank you for highlighting the ambiguity in the description of our auditory stimulus delivery setup. We have revised the corresponding section in the manuscript to provide a clearer explanation of how participants received the auditory stimuli. The sound was delivered binaurally using loudspeakers located outside the MEG's shielded room and transmitted to the participants inside the chamber through a specialized sound-conduction system that took advantage of a gap in the chamber's design. This ensured that the quality of the sound remained intact without any interference with the MEG measurements. We hope this provides a clearer picture of our setup.

(Auditory stimuli and procedures)

Participants received the auditory stimuli binaurally, meaning through both ears. The stimuli were transmitted via loudspeakers (HK195 Speakers; Harman Kardon, Stamford, CT, USA) located outside the magnetically shielded room housing the MEG equipment. The speakers delivered the sound into the MEG chamber through a specialized sound-conduction system that utilized a gap or aperture in the chamber's structure, ensuring the sound quality was maintained without interfering with the MEG's magnetic field. This setup facilitated a 12-minute stimulus-presentation session.

Lines 341-343: ‘Furthermore, in contrast to the ASD group, we did not find a significant correlation between the right P1m intensity and SRS total T-score in TD children, possibly due to a lack of power in our sample (53).’

Why the authors expected to find such correlation?

Thank you for your query regarding the expected correlation between right P1m intensity and SRS total T-score in TD children. Upon reanalyzing our data and revising the Results and Discussion sections, we realized that this particular statement may not align precisely with our primary findings and hypotheses. Consequently, we have removed this section from the manuscript to ensure clarity and coherence. We appreciate your feedback, as it has guided us in refining the presentation of our results.

The study would benefit from help of English editing services.

e.g. in abstract:

‘On performing multiple regression analyses using P1m intensity in the right and left hemispheres and the K-ABC Mental Processing Scale score as the dependent variables, and using the SRS total T-score as the independent variable, we identified right P1m intensity as a predictor of the SRS total T-score in children with ASD, and this relationship was not found in TD children. ’

Thank you for your constructive feedback regarding the clarity and language of our manuscript. We acknowledge the importance of presenting our findings in clear and coherent English. In response to your suggestion, we have sought the expertise of a native English proofreader to review and refine the language throughout our manuscript. We trust that this step has enhanced the clarity and readability of our work. We appreciate your patience and understanding in this matter.

6. PLOS authors have the option to publish the peer review history of their article (what does this mean?). If published, this will include your full peer review and any attached files.

Do you want your identity to be public for this peer review? For information about this choice, including consent withdrawal, please see our Privacy Policy.

Reviewer #1: Yes: Wan-Chun Su

Reviewer #2: No

---

## [Decision Letter · Decision Letter 1]

20 Dec 2023

PONE-D-23-19599R1Neural Responses to Syllable-Induced P1m and Social Impairment in Children with Autism Spectrum Disorder and Typically Developing PeersPLOS ONE

Dear Dr. Hirosawa,

Thank you for submitting your manuscript to PLOS ONE. After careful consideration, we feel that it has merit but does not fully meet PLOS ONE’s publication criteria as it currently stands. Therefore, we invite you to submit a revised version of the manuscript that addresses the points raised during the review process.

**l**Thank you for your thoughtful edits and for considering and addressing our concerns.Please address the remaining concerns with the same efforts and endeavour.Wishing you success with the study.

We look forward to receiving your revised manuscript.

Kind regards,

Thiago P. Fernandes, PhD

Academic Editor

PLOS ONE

Reviewers' comments:

Reviewer's Responses to Questions

**Comments to the Author**

1. If the authors have adequately addressed your comments raised in a previous round of review and you feel that this manuscript is now acceptable for publication, you may indicate that here to bypass the “Comments to the Author” section, enter your conflict of interest statement in the “Confidential to Editor” section, and submit your "Accept" recommendation.

Reviewer #2: (No Response)

2. Is the manuscript technically sound, and do the data support the conclusions?

Reviewer #2: Partly

3. Has the statistical analysis been performed appropriately and rigorously? 

Reviewer #2: Yes

4. Have the authors made all data underlying the findings in their manuscript fully available?

Reviewer #2: Yes

5. Is the manuscript presented in an intelligible fashion and written in standard English?

Reviewer #2: Yes

6. Review Comments to the Author

Reviewer #2: I appreciate the revisions made by the authors. They discovered an error in data processing and reanalyzed the data. In fact, the entire text has changed, including results, discussion, and even the title of the manuscript. However, I still have a few comments and/or suggestions:

Major

1. The authors noted that in the ‘limitations’ that ‘… limitation arises from setting thebaseline relative to the onset of the vowel /e/. Brain responses to the preceding consonant /n/ may exist, and the proportion of participants demonstrating this response might differ between the TD and ASD groups, potentially influencing the results. To validate this, future research should include both /ne/ and /e/ stimuli.’

In response to the reviewer #2 they also noted that ‘In the first manuscript, the baseline was mistakenly set to −50 ms to 0 ms relative to the onset of the /n/ consonant’.

This means that in the previous study they analyzed P1m in the time range that overlapped with the range of baselines used in the present study. Why do they believe that the results obtained with a more neutral baseline (i.e., before the presentation of the syllable /ne/) are less correct than the present results where the baseline overlaps with the presentation of the consonant /n/? Arguments and illustrations are needed to support the choice of baseline.

2. The authors included in the introduction a paragraph about ‘P1m and "suppression" phenomenon’. However, they did not investigate the suppression (gating) in their study. Moreover, gating is usually investigated using short non–speech stimuli, such as clicks, while Hirosawa et al used speech stimuli. Therefore, I am not sure that this paragraph is relevant for the discussion.

On the other hand, there is literature that may be more relevant. Indeed, P1 abnormalities were reported in the auditory processing disorder (APD). P1 amplitude evoked by a speech stimulus (/da/) has been shown to be smaller in children with APD compared to TD control children (Sharma et al, 2014; DOI: http://dx.doi.org/10.1055/s-0033-1363524). Another recent study also found P1 abnormalities in APD in response to speech stimuli (doi: 10.1016/j.ijporl.2021.110944). As many children with ASD may have APD (James, 2022; https://doi.org/10.1016/j.jcomdis.2022.106252), P1 finding of Hirosawa et al may, at least to some expend, reflect central auditory processing deficit, rather than ASD itself. APD and receptive language were not assessed in the present study. It seems to be an important limitation, which needs to be discussed.

3. In p. 14, the authors report no group differences in P1 amplitude and latency. Does the asymmetry of P1 amplitude differ between TD and ASD? (I apologize if I missed this information. Perhaps it would have been better to report it directly after the group differences in latency and amplitude).

4. Thank you for your response to comment 8. You write: ‘However, this study did not account for the impact of fine head movements and variations in head shape.’

This is not exactly what my concern was about. The reviewer is aware of a case in which the group mean head positions of subjects from two different clinical groups differed statistically significantly in the left-to-right direction during MEG recording. Even if you excluded subjects who moved a lot, the left-right position by itself (not ‘fine head movements’ or ‘head shape’) could differ between ASD and TD participants, which in turn might affect the results. Once you have measured the head position of each subject at least once, you could probably report information about the differences between ASD and TD (left-to-right position, rotation), and also discuss the limitations (not tracking head position).

It would also be interesting to know whether L-R P1m lateralization correlates with this [baseline] position.

5. Fig 1. Waveform of the standard /ne/ (left) and deviant /Ne/ (right) voice stimulus. The total duration of each stimulus was 342 ms, segmented into 65 ms for the consonant /n/ and 277 ms for the subsequent vowel sound /e/. The MEG analysis onset time was defined as the beginning of the vowel portion. It is important to note that only the standard stimuli were used for the subsequent equivalent current dipole (ECD) estimation, as only 328 this condition provided a sufficient number of epochs for accurate ECD calculation.

From this description it is unclear what is presented in figure 1. Is it a representative subject or the group average? Please provide figures for this time interval separately for TD and ASD subjects (each group average) in the left and right hemispheres.

Discussion.

6. The P1m results are interesting to discuss in relation to the possible contribution of the sustained negative shift of current, which has been described in adults ( Gutschalk and Uppenkamp, 2011; doi:10.1016/j.neuroimage.2011.02.026) and, more recently ,in both adults and children (Orekhova et al, 2023; https://doi.org/10.1016/j.cortex.2023.10.020). Figure 3 in Orekhova et al. shows that the amplitude and possibly the latency of P1 (=P100m) can be influenced by a sustained negative shift associated with the processing of phonetic features of the speech stimulus. The fact that different physiological processes may contribute to auditory ERP/ERF activity over the P1m time range allows for different interpretations of the P1m results.

7. The correlation between P1m latency and SRS was observed in TD group, but not in the ASD. The authors suggested that ‘.. rather than directly indicating autistic traits, a shorter latency of the left P1m might represent a neural adaptation against these traits’.

However, it is not clear if SRS scale reflects the same underlying physiology in TD and ASD. Indeed, in ASD it may reflect serious deficit, such as a degree of E/I imbalance, while it would be strange to expect presence of such deficit in TD.

It's up to the authors to decide, but in my opinion, their interpretation looks a bit naive.

Minor

1. Thank you for your detailed response to my previous comment 2. I think this information may be interesting mot only for the reviewer. Therefore, I suggest to summarize briefly the main results in the text (e.g. ‘the correlation was reproduced even for a smaller sample not included in the previous study’) and to address the reader to the S6.

2. Lines 591-

(1)We identified a notable association between a shorter latency of syllable-induced P1m in the left hemisphere and pronounced autistic traits. Interestingly, this correlation is primarily evident in TD children and appears nonsignificant in children with ASD. (2) At a glance, one might infer that a shorter latency of the left P1m serves as a neural marker for prominent autistic traits overall. (3) If this assumption is accurate, then a shorter latency of the left P1m could potentially be linked with lower conceptual inference skills, an identified facet of autistic symptomatology [62].

It is not clear how (3) follows directly from (1 and 2), since conceptual inference skills may reflect IQ rather than the autism disorder itself. According to this logic, one would expect a correlation between P1 latency and IQ/MPS in both groups.

3. Some tables are very difficult to read because it is difficult to understand what raw corresponds to what, e.g.:

(see in attached file)

4. Fig. 4. Some information in the legend is redundant (e.g. ‘The horizontal axis showcases the latency of P1m in the left hemisphere, ranging from 40 to 120 ms.’, etc.) This one can see on the plot. On the other hand, there are no individual data points, which were present in the previous version. Please show individual data points in figures 4 and 5 (different marks for ASD and TD children). I would suggest removing bars and marking confidence intervals with lines.

5. Figures 5. It may be convenient for the reader see directly on the plot what is the ‘leftward lateralization’, e.g.:

R>L R<l< p=""></l<>

7. PLOS authors have the option to publish the peer review history of their article (what does this mean?). If published, this will include your full peer review and any attached files.

Reviewer #2: No

---

## [Author Response · Author response to Decision Letter 1]

16 Jan 2024

I appreciate the revisions made by the authors. They discovered an error in data processing and reanalyzed the data. In fact, the entire text has changed, including results, discussion, and even the title of the manuscript. However, I still have a few comments and/or suggestions:

Major

1. The authors noted that in the ‘limitations’ that ‘… limitation arises from setting thebaseline relative to the onset of the vowel /e/. Brain responses to the preceding consonant /n/ may exist, and the proportion of participants demonstrating this response might differ between the TD and ASD groups, potentially influencing the results. To validate this, future research should include both /ne/ and /e/ stimuli.’

Response: ‘In the first manuscript, the baseline was mistakenly set to −50 ms to 0 ms relative to the onset of the /n/ consonant’.

This means that in the previous study they analyzed P1m in the time range that overlapped with the range of baselines used in the present study. Why do they believe that the results obtained with a more neutral baseline (i.e., before the presentation of the syllable /ne/) are less correct than the present results where the baseline overlaps with the presentation of the consonant /n/? Arguments and illustrations are needed to support the choice of baseline.

Method Section Revision:

Response: Thank you for your insightful comment. Per your suggestion, we have refined our manuscript as follows:

We acknowledge the significance of selecting the appropriate onset time for syllable stimuli, specifically between the consonant /n/ and the vowel /e/. To maintain consistency with our previous studies [29, 56-58, 62], which predominantly examined the vowel /e/ response, we opted for this latter setting in our current study. This choice is pivotal for enabling direct comparisons with prior findings, thereby enriching our understanding of auditory processing in typically developing children and children with ASD. However, we must acknowledge a crucial assumption in this approach: a minimal brain response to the consonant /n/ due to its lower sound intensity compared to the vowel /e/. While this assumption would be reasonable, it may not fully encapsulate the natural auditory processing mechanisms and could inadvertently obscure the brain's response to the consonant /n/. This aspect warrants careful consideration in interpreting our findings.

Discussion Section Revision:

We have also updated the Discussion section to address potential methodological limitations:

Our study's focus on the vowel /e/, setting the baseline relative to its onset, inherently implies a possible oversight of the brain's response to the preceding consonant /n/. This approach, while methodologically sound for our current research objectives, may mask nuances in the brain's processing of the /n/ consonant. This limitation is particularly pertinent when considering the differential responses between TD children and children with ASD. Our findings, thus, should be interpreted with an awareness of this potential masking effect. In light of this, future research endeavors should contemplate including stimuli combinations like /ne/ and /e/ to comprehensively investigate the brain's distinct responses to consonants and vowels. Such explorations would be instrumental in deepening our understanding of auditory processing variations between TD and ASD groups, potentially leading to more nuanced insights into their auditory processing characteristics.

We hope these revisions effectively address your concerns and clarify the rationale behind our methodological choices, thereby enhancing the overall quality and integrity of our research. 

2. The authors included in the introduction a paragraph about ‘P1m and "suppression" phenomenon’. However, they did not investigate the suppression (gating) in their study. Moreover, gating is usually investigated using short non–speech stimuli, such as clicks, while Hirosawa et al used speech stimuli. Therefore, I am not sure that this paragraph is relevant for the discussion.

On the other hand, there is literature that may be more relevant. Indeed, P1 abnormalities were reported in the auditory processing disorder (APD). P1 amplitude evoked by a speech stimulus (/da/) has been shown to be smaller in children with APD compared to TD control children (Sharma et al, 2014). Another recent study also found P1 abnormalities in APD in response to speech stimuli (Lunardelo, 2021). As many children with ASD may have APD (James, 2022; https://doi.org/10.1016/j.jcomdis.2022.106252), P1 finding of Hirosawa et al may, at least to some expend, reflect central auditory processing deficit, rather than ASD itself. APD and receptive language were not assessed in the present study. It seems to be an important limitation, which needs to be discussed.

Response: Per the reviewer's insightful comment, we have made the following amendments to our manuscript:

We have removed the paragraph in the Introduction that discussed P1m and the suppression phenomenon in the context of psychopathologies. This adjustment aligns the focus of our introduction more closely with the scope of our study.

In the Discussion section, particularly in the Limitations subsection, we have incorporated the reviewer's suggestions as follows:

Limitations Section Revision:

Another limitation, as underscored by recent research, concerns the potential intersection of Auditory Processing Disorder (APD) symptoms within the ASD population. Studies by Sharma et al. [1] and Lunardelo et al. [2] revealed P1 amplitude abnormalities in children with APD in response to speech stimuli, specifically the /da/ sound, which is similar to the /ne/ sound utilized in our research. These findings imply that P1 irregularities may not be unique to ASD and could also signify a central auditory processing deficit characteristic of APD. Moreover, the work of James et al. [3] indicates a potentially high incidence of APD symptoms among children with ASD. This overlap suggests that the P1 abnormalities we observed in children with ASD might partially reflect a broader spectrum of auditory processing challenges extending beyond the confines of ASD. The absence of a direct evaluation of APD in our study is a notable oversight. This limitation warrants caution in attributing the P1 abnormalities solely to ASD and suggests the need for future research to disentangle the auditory processing profiles of ASD from those of APD. Undertaking such research would provide a more comprehensive understanding of the auditory processing dynamics in neurodevelopmental disorders.

 

3. In p. 14, the authors report no group differences in P1 amplitude and latency. Does the asymmetry of P1 amplitude differ between TD and ASD? (I apologize if I missed this information. Perhaps it would have been better to report it directly after the group differences in latency and amplitude).

Response: Thank you for your insightful comment regarding the presentation of P1 amplitude asymmetry data in our manuscript. We appreciate your attention to this detail and understand the importance of clearly communicating these findings.

In our original manuscript, we reported the absence of significant group differences in P1 amplitude and latency on page 14. We agree with your suggestion that information about the asymmetry of P1 amplitude is also of interest to readers. However, we chose to detail the asymmetry analysis in a later section (More pronounced autistic symptoms are associated with stronger leftward lateralization in P1m intensity, exclusively in children with ASD) due to its methodological complexity, including steps like outlier identification and exclusion.

To address your concern and enhance the clarity of our manuscript, we have now included a brief mention of the P1 amplitude asymmetry results in the section discussing group differences in P1 amplitude and latency. Specifically, we have added the following statement on page 14:

“Besides reporting no significant group differences in P1 latency and log-transformed intensity, we also examined the asymmetry of P1 intensity between the TD and ASD groups. It is important to note that this analysis revealed no significant difference in the asymmetry of P1 amplitude between the two groups. We delve into the details of this analysis, including the calculation method and considerations for outlier exclusion, in a later section of this manuscript.”

This addition aims to provide a concise overview of the P1 amplitude asymmetry findings while directing readers to the section where the analysis is discussed in depth. We believe this revision addresses your concern and improves the manuscript by making this information more accessible while maintaining the logical flow and thoroughness of our analysis.

We hope this modification satisfactorily addresses your comment, and thank you again for your constructive feedback. 

4. Thank you for your response to comment 8. You write: ‘However, this study did not account for the impact of fine head movements and variations in head shape.’ 

This is not exactly what my concern was about. The reviewer is aware of a case in which the group mean head positions of subjects from two different clinical groups differed statistically significantly in the left-to-right direction during MEG recording. Even if you excluded subjects who moved a lot, the left-right position by itself (not ‘fine head movements’ or ‘head shape’) could differ between ASD and TD participants, which in turn might affect the results. Once you have measured the head position of each subject at least once, you could probably report information about the differences between ASD and TD (left-to-right position, rotation), and also discuss the limitations (not tracking head position). 

It would also be interesting to know whether L-R P1m lateralization correlates with this [baseline] position.

Response: Thank you for your valuable feedback regarding the impact of initial head positions on our P1m results. Based on your comment, we have made significant revisions to our manuscript to address these concerns. 

In the Results section, we have included a detailed analysis of the initial head positions of the participants, attaching three coils to each subject's skull at both mastoid processes and the nasion. This allowed us to track their initial head positions accurately. Our analysis revealed significant differences between the ASD and TD groups in the y-coordinate of the coil at both the left and right mastoid processes. These findings, which indicate a more posterior position of the mastoid processes in children with ASD compared to TD children, are now thoroughly reported and discussed.

Furthermore, we investigated the correlation between the leftward lateralization in log-transformed P1m intensity and the initial head positions. A significant correlation was found between the leftward lateralization and the x-coordinate of the coil at the nasion, suggesting that a more left-located coil at the nasion corresponds to a larger leftward lateralization of intensity. This correlation, along with the lack of significant associations in other models, is detailed in Supplementary Table 7.

In the Discussion section, particularly under limitations, we acknowledge that despite efforts to monitor and control head position, the significant differences in initial head positions could potentially influence the results of dipole estimation. We discuss the implications of these findings and the need for future research to more thoroughly consider the impact of fine head movements and variations in head shape.

These revisions aim to comprehensively address your concerns about the role of the initial head position in interpreting our P1m results. We believe that these additions enhance the rigor and clarity of our study, providing a more accurate representation of the potential variables influencing our findings.

Thank you again for your constructive feedback, which has been instrumental in strengthening our manuscript.

Result section revision

To ensure that the initial head positions did not differ statistically between the ASD and TD groups, we attached three coils to each subject's skull, positioned at both mastoid processes and the nasion. Each coil created a magnetic field that enabled us to track their initial head positions. A Student’s t-test revealed significant differences between the ASD and TD groups in the y-coordinate of the coil at both the left mastoid process (t(62) = -2.22, p = 0.03) and the right mastoid process (t(62) = -2.05, p = 0.04), which might affect the results (as discussed in the limitations section). No significant differences were observed in the x and z coordinates of these coils. Similarly, no significant differences were found in any coordinate of the coil at the nasion. Detailed results are presented in Supplementary Table 1.

Results section revision

As significant differences were observed in the initial head position (i.e., y-coordinate of the coils at both the left and right mastoid process), we investigated whether the leftward lateralization in log-transformed P1m intensity correlated with these initial head positions. To this end, we employed simple regression analysis to predict the leftward lateralization in log-transformed P1m intensity based on the x, y, or z coordinates of each coil separately. A significant correlation was found between the leftward lateralization in log-transformed P1m intensity and the x coordinate of the coil at the nasion (t(50) = -2.61, p = 0.01), indicating that a larger leftward lateralization of intensity corresponds to a left-located coil at the nasion. No significant associations were observed in any of the other models. Detailed results are presented in Supplementary Table 5.

Discussion section (limitation) revision

In this study, participants were monitored using a video camera to detect noticeable body movements. An examiner accompanied the children in the shielded room and instructed them to maintain a constant head position throughout the experiment. Instances of pronounced body movement were excluded based on noise detection. Additionally, participants who exhibited significant shifts in head position during the session were excluded due to a reduction in the GOF in the P1m dipole analysis. Despite these measures, we observed significant differences in the initial head positions between the two groups. Specifically, the positions of both the right and left mastoid processes in children with ASD were significantly more posterior compared to those in TD children. This difference could reflect variations in initial head positioning or head shape; either factor could potentially influence the results of dipole estimation. Indeed, while the position of the mastoid process did not affect the leftward lateralization of log-transformed P1m intensity, the x-coordinate of the coil at the nasion was found to significantly influence the estimation of this parameter. Furthermore, this study did not comprehensively account for the impact of fine head movements and variations in head shape, which are factors that could introduce additional variability in the neuroimaging data. Future research should consider these aspects more thoroughly to mitigate their potential effects on data interpretation.

Supplementary Table 1:

Differences in x, y, and z coordinates of the head coils. Larger (smaller) values of the x coordinate correspond to the left (right) direction, respectively. Larger (smaller) values of the y coordinate correspond to the posterior (anterior) direction, respectively. Larger (smaller) values of the z coordinate correspond to the head (foot) direction, respectively.

Supplementary Table 5.

Correlation between leftward lateralization in log-transformed P1m intensity and initial head positions

 

5. Fig 1. Waveform of the standard /ne/ (left) and deviant /Ne/ (right) voice stimulus. The total duration of each stimulus was 342 ms, segmented into 65 ms for the consonant /n/ and 277 ms for the subsequent vowel sound /e/. The MEG analysis onset time was defined as the beginning of the vowel portion. It is important to note that only the standard stimuli were used for the subsequent equivalent current dipole (ECD) estimation, as only 328 this condition provided a sufficient number of epochs for accurate ECD calculation.

From this description it is unclear what is presented in figure 1. Is it a representative subject or the group average? Please provide figures for this time interval separately for TD and ASD subjects (each group average) in the left and right hemispheres.

Response: Thank you for your valuable feedback regarding the visual representation of our data in the manuscript. We understand your interest in seeing the group averages for the TD and ASD participants in both hemispheres.

To clarify, Figure 1 in our manuscript is intended to depict the waveform of the auditory stimuli (standard /ne/ and deviant /Ne/) used in our study. Its purpose is to provide a clear understanding of the stimuli's structure and duration, which is foundational for our experiment. The figure shows the sound waveforms, demonstrating the segmentation of the consonant /n/ and the vowel /e/.

Additionally, we have noted your interest in viewing the MEG response data for both TD and ASD participants. While we originally presented a representative subject's neural response in Figure 2, we acknowledge the value of your suggestion. To address this, we have decided to include new Supplementary Figure 1 that displays the group averages for the TD and ASD participants in the left and right hemispheres. These figures will provide a comprehensive view of the neural response patterns across both groups and hemispheres, adding depth to our analysis and discussion.

We believe these additions, along with the clarification of Figure 1's purpose, will enhance the understanding and impact of our study. We hope this modification meets your requirements and further strengthens our manuscript.

Now legends of Fig. 1 and Fig. 2 read as follows:

Fig 1. Waveform of the Auditory Stimuli.

This figure presents the sound waveforms of the standard /ne/ (left panel) and deviant /Ne/ (right panel) voice stimuli used in the study. The total duration of each stimulus is 342 ms, segmented into 65 ms for the consonant /n/ and 277 ms for the subsequent vowel sound /e/. This illustration is intended to provide a clear understanding of the structural and temporal characteristics of the stimuli employed in our experiments. The MEG analysis onset time was defined as the beginning of the vowel portion. It is important to note that only the standard stimuli were used for the subsequent equivalent current dipole (ECD) estimation, as only this condition provided a sufficient number of epochs for accurate ECD calculation.

Fig 2. Neuromagnetic response to the standard syllable /ne/ stimuli.

This figure presents these waveforms and the magnetic contour map of P1m for a representative participant. Syllable-induced AEF with a baseline from −50 to 0 ms relative to the onset of the vowel /e/. The resultant AEF displayed a pronounced activity peak between 45 and 150 ms. The onset of the consonant /n/ is at −65 ms relative to that of /e/. The blue arrow displays the direction of the estimated dipole moment.

S1_Fig. Neuromagnetic response to the standard syllable /ne/ stimuli for typical development（TD）and autism spectrum disorder（ASD）participants (each group average) in the left and right hemispheres. Syllable-induced AEF with a baseline from −50 to 0 ms relative to the onset of the vowel /e/. The onset of the consonant /n/ is at −65 ms relative to that of /e/. 

Discussion.

6. The P1m results are interesting to discuss in relation to the possible contribution of the sustained negative shift of current, which has been described in adults ( Gutschalk and Uppenkamp, 2011; doi:10.1016/j.neuroimage.2011.02.026) and, more recently, in both adults and children (Orekhova et al, 2023; https://doi.org/10.1016/j.cortex.2023.10.020). Figure 3 in Orekhova et al. shows that the amplitude and possibly the latency of P1 (=P100m) can be influenced by a sustained negative shift associated with the processing of phonetic features of the speech stimulus. The fact that different physiological processes may contribute to auditory ERP/ERF activity over the P1m time range allows for different interpretations of the P1m results. 

Response: Thank you for your insightful comment regarding the interpretation of our P1m results, particularly in relation to the potential contribution of the sustained negative shift of current, as described in the studies by Gutschalk and Uppenkamp (2011) and Orekhova et al. (2023). Your observation about the possible influence of this sustained negative shift on both the amplitude and latency of the P1m response is indeed a significant consideration for our study.

In response to your comment, we have revised our manuscript to incorporate a more detailed discussion. We have explored how the sustained field (SF), associated with the processing of periodicity/pitch and formant structure in speech stimuli, might interact with and influence the P1m component. This SF, as evidenced in the literature, is thought to reflect the activity of non-synchronized neuronal populations in the auditory cortex, which serve as feature detectors for complex sounds. The potential for this SF to be present in the time range of the P1m component or even earlier suggests that our observed association between autistic traits and syllable-induced P1m latency might also encompass the processing of these perceptually salient features of speech stimuli.

Additionally, we have considered the implications of these findings for understanding auditory processing in children with ASD. The emerging evidence points to atypical attentional responses to speech and higher-level processing anomalies in this population, which could be intertwined with the dynamics of the sustained negative shift.

Our revised discussion section now offers a richer interpretation of the P1m results, considering the complexity of auditory processing and the potential co-occurrence of early auditory responses with more elaborate phonetic feature processing. We believe this enhanced discussion addresses your concern and adds depth to our analysis, aligning with the evolving understanding of auditory processing mechanisms in neurodevelopmental disorders like ASD.

We appreciate your valuable feedback, which has significantly contributed to the improvement of our manuscript.

Discussion section revision:

Another approach to interpreting our P1m results involves considering the broader framework of auditory processing, particularly in relation to the potential contribution of the 'sustained negative shift' of current, as described in adults [4] and in both adults and children [5]. The processing of sounds characterized by periodicity/pitch and/or formant structure, such as vowels, is associated with a greater sustained negative shift of cortical source current, known as the sustained field (SF), which persists throughout stimulus presentation. This SF, captured by MEG/EEG, is thought to reflect the activation of non-synchronized neuronal populations [13,14]. These neurons function as 'feature detectors' for perceptually salient features of complex sounds, facilitating higher-level processing [15,16,17]. The enhancement of MEG/EEG-measured SF occurs when stimuli are perceptually salient [7] or carry semantic meaning [6], and its magnitude varies with phonetic features, such as periodic versus non-periodic vowels [5]. Notably, SF is evident in the time range of the P1m component or even earlier, suggesting that the co-occurrence of SF with P1m might influence the contour and amplitude of the P1m. This interaction is particularly relevant in our findings, where we observed an association between autistic traits and syllable-induced P1m latency and its leftwared lateralization in intensity, possibly reflecting a latent relationship between autistic traits and SF. The potential connection between autistic traits and SF is compelling, given the emerging behavioral and electrophysiological evidence of impaired attentional responses to speech in children with ASD. This might imply reduced perceptual salience of speech stimuli and atypical higher-level processing. Earlier studies indicate that children with ASD exhibit specific deficits in orienting to vowel sounds compared to simple and complex tones [8], highlighting their potentially reduced perceptual salience to speech. Moreover, these deficits may be linked to atypical higher-level processing of auditory stimuli in this population [9]. Given these considerations, future research aiming to further elucidate the complex interplay between autistic traits, SF, and properties of P1m could offer a richer understanding of the neural basis of ASD and how it is reflected in MEG/EEG measurements.

7. The correlation between P1m latency and SRS was observed in TD group, but not in the ASD. The authors suggested that ‘.. rather than directly indicating autistic traits, a shorter latency of the left P1m might represent a neural adaptation against these traits’.

However, it is not clear if SRS scale reflects the same underlying physiology in TD and ASD. Indeed, in ASD it may reflect serious deficit, such as a degree of E/I imbalance, while it would be strange to expect presence of such deficit in TD. It's up to the authors to decide, but in my opinion, their interpretation looks a bit naive.

Response: Thank you for your insightful comments regarding the interpretation of the correlation between P1m latency and SRS scores in our study. We appreciate your observation that the SRS scale may not reflect the same underlying physiological phenomena in TD and ASD individuals. In response to your feedback, we have revised our manuscript to offer a more nuanced interpretation of these findings.

In the revised section, we acknowledge that the SRS in TD children could represent a range of cognitive processing styles related to autistic traits, while in ASD children, it might reflect more specific aspects of ASD pathology, such as excitatory/inhibitory (E/I) imbalances. This distinction is important to consider, as it suggests that the shorter P1m latency observed in TD children could be indicative of a broad spectrum of cognitive processing, not necessarily direct markers of autistic pathology. Conversely, the lack of a significant correlation in ASD children suggests the involvement of different neural processes, potentially linked to specific neurophysiological characteristics characteristic of ASD.

We also address the possibility that the shorter P1m latency in TD children might reflect neural processes less directly related to autistic pathology, perhaps involved in more general social information processing. This interpretation contrasts with the situation in children with ASD, where the pronounced social deficits might be more directly related to autism-specific pathology, which might not be reflected in P1m latency.

Our revision aims to provide a more comprehensive understanding of the complex interplay between auditory processing and social responsiveness traits in both populations. We believe this refined interpretation aligns better with the complexities of neurodevelopmental differences between TD and ASD individuals.

We hope this revised interpretation addresses your concerns and provides a more accurate reflection of the nuanced relationships between neural markers, SRS scores, and the distinct neurodevelopmental contexts of TD and ASD.

Discussion revision:

We identified a significant association between a shorter latency of syllable-induced P1m in the left hemisphere and pronounced autistic traits. Interestingly, this correlation is primarily evident in TD children and appears nonsignificant in children with ASD. At first glance, this finding suggests a potential link between neural auditory processing and autistic traits. However, it is crucial to consider that the SRS scale may not reflect the same underlying physiology in TD and ASD individuals. In TD children, the SRS could represent a range of cognitive processing styles related to autistic traits, while in ASD children, it might reflect more specific aspects of ASD pathology, such as excitatory/inhibitory (E/I) imbalances [18].

From this perspective, the observed correlation between shorter P1m latency and higher SRS scores in TD children could indicate a relationship between P1m latency and a broad spectrum of cognitive processing that relates to autistic traits rather than being solely indicative of autism-specific pathology. This spectrum might include neural adaptations or processing efficiencies unrelated to autism but still captured by SRS scores.

Conversely, the lack of a significant correlation in ASD children hints at the involvement of different neural processes that are reflected in their SRS scores. These processes could be linked to specific neurophysiological characteristics, such as E/I imbalances, which are considered characteristic of ASD pathology [18].

Therefore, the shorter P1m latency in TD children might reflect a neural process less directly related to autistic pathology, perhaps involved in more general social information processing. In contrast, in children with ASD, given their pronounced social deficits, such general social information processing might no longer be associated with the severity of their social deficits. Instead, their social deficits might be more directly related to autism-specific pathology, which might not be reflected in P1m latency. This could explain why we failed to find a significant relation between P1m latency and SRS scores in this population.

Our findings highlight the need for further research to explore the specific neurophysiological underpinnings of SRS scores in both TD and ASD individuals. Future studies should aim to disentangle the relationships between neural markers like P1m latency, autism-specific pathology, and other mechanisms involved in social information processing. Such research would offer a clearer understanding of the complex interplay between auditory processing and social responsiveness traits in both populations. 

Minor

1. Thank you for your detailed response to my previous comment 2. I think this information may be interesting mot only for the reviewer. Therefore, I suggest to summarize briefly the main results in the text (e.g. ‘the correlation was reproduced even for a smaller sample not included in the previous study’) and to address the reader to the S6. 

Response: Thank you for your appropriate advice. Following your advice, we have added the following to the Discussion section.

Discussion Section Revision:

We have added the following part at the end of the Discussion section:

Results of analyses with new participants only

In the present study, some participants overlapped with participants included in our previous studies. We excluded them and performed all analyses on 'new' participants only to test whether the results could be reproduced. Twenty-eight children with ASD and 18 TD children were included in these analyses. To summarise the main results, the relationship between left P1m latency and SRS T-scores in the TD group did not maintain statistical significance; the association between SRS total T-score and leftward lateralization in P1m log-intensity in the ASD group remained statistically significant. Other detailed results are given in the S8 tables.

2. Lines 591-

(1)We identified a notable association between a shorter latency of syllable-induced P1m in the left hemisphere and pronounced autistic traits. Interestingly, this correlation is primarily evident in TD children and appears nonsignificant in children with ASD. (2) At a glance, one might infer that a shorter latency of the left P1m serves as a neural marker for prominent autistic traits overall. (3) If this assumption is accurate, then a shorter latency of the left P1m could potentially be linked with lower conceptual inference skills, an identified facet of autistic symptomatology [62].

It is not clear how (3) follows directly from (1 and 2), since conceptual inference skills may reflect IQ rather than the autism disorder itself. According to this logic, one would expect a correlation between P1 latency and IQ/MPS in both groups.

Response: We have removed that section of the manuscript as it was entirely restructured in response to major comment 7.

3. Some tables are very difficult to read because it is difficult to understand what raw corresponds to what, e.g.:

Response: We apologize for the error. We will ensure that such errors do not occur during this submission.

4. Fig. 4. Some information in the legend is redundant (e.g. ‘The horizontal axis showcases the latency of P1m in the left hemisphere, ranging from 40 to 120 ms.’, etc.) This one can see on the plot. On the other hand, there are no individual data points, which were present in the previous version. Please show individual data points in figures 4 and 5 (different marks for ASD and TD children). I would suggest removing bars and marking confidence intervals with lines.

Figures 5. It may be convenient for the reader see directly on the plot what is the ‘leftward lateralization’, e.g.: 

R>L R<L

Response: Thank you for your valuable feedback regarding Figures 4 and 5 in our manuscript. We appreciate your suggestions for improving the clarity and informativeness of these figures.

Regarding your comment on Figure 4, we have revised the legend to eliminate redundant information. As you rightly pointed out, certain details, such as the range of P1m latency, are directly observable from the plot and thus do not need to be reiterated in the legend. Additionally, in response to your suggestion, we have now included individual data points in the figure. These points provide a more detailed and informative visualization.

For Figure 5, we have taken steps to directly illustrate the concept of 'leftward lateralization' in the plot itself, making it more accessible and understandable for readers. This change should facilitate a more immediate comprehension of the relationship between SRS total T-scores and P1m's leftward lateralization in log-transformed intensity. We have also ensured that individual data points for both ASD and TD children are clearly represented, with different markers for each group.

We have replaced the bars with lines to mark the confidence intervals, in both figures. This modification not only enhances the visual appeal of the figures but also aids in better interpretation of the confidence intervals.

We believe these changes address your concerns effectively and enhance the overall quality and clarity of the figures. We hope that these revisions meet your approval and improve the manuscript's ability to communicate the results clearly to its readers.

Figure 4: Relationship Between SRS Total T-Scores and P1m Latency in the Left Hemisphere

To visualize the relation between SRS total T-scores and P1m latency in the left hemisphere for TD children, we performed a simple regression to predict SRS total T-scores based solely on P1m latency, excluding the mental processing scale for clarity. The effect of P1m latency on SRS total T-scores remains significant in this simplified model (t(19) = -2.68, p = 0.015). The figure depicts individual data points for TD children. The solid line represents the predicted regression line, and the shaded area around it denotes the 95% confidence intervals based on our regression model.

SRS, social responsiveness scale; TD children, typically developing children.

Figure 5: Relationship Between SRS Total T-Scores and P1m's Leftward Lateralization

This figure illustrates the relationship between SRS total T-scores and leftward lateralization in log-transformed intensity (defined as the log-transformed P1m intensity in the left hemisphere minus that in the right) for both ASD and TD children. Separate simple regressions were performed for each group to predict SRS total T-scores based on this measure of P1m's leftward lateralization, excluding the mental processing scale for clarity. The effect of leftward lateralization in log-transformed P1m intensity on SRS total T-scores was found to be significant only for TD children (t(28) = 2.15, p = 0.04). Individual data points for each group are plotted, with each line corresponding to a diagnostic group, illustrating how predicted SRS scores vary with P1m’s leftward lateralization.

SRS, social responsiveness scale; ASD, autism spectrum disorder; TD children, typically developing children.

---

## [Editor Report · Decision Letter 2]

17 Jan 2024

Neural Responses to Syllable-Induced P1m and Social Impairment in Children with Autism Spectrum Disorder and Typically Developing Peers

PONE-D-23-19599R2

Dear Dr. Hirosawa,

We’re pleased to inform you that your manuscript has been judged scientifically suitable for publication and will be formally accepted for publication once it meets all outstanding technical requirements.

Kind regards,

Thiago P. Fernandes, PhD

Academic Editor

PLOS ONE

Additional Editor Comments (optional):

After a careful re-read, I think the authors did a good job in addressing the concerns raised. From my standpoint, it's commendable for transparency re. the identified error, especially with tweaks in structure and interpretation. Nevertheless, merit shouldn't hinge solely on this; instead, it should encompass the communication, soundness, and the findings. Hence, the study retains its potential, given that the differences were not overstated.

On the top on that, it is essential to express appreciation for the insightful comments from Rev #2, which have greatly contributed to enhancing the study's clarity and robustness. 

A reminder: keep the data on OSF, double-check refs. and grammar and maintain consistency in presentation of data. Also, if there are any additions deemed useful that could be placed as Sup. files, during typesetting, consider it, as this can help readers and researchers.

Wishing you success with the study.
---

## [Editor Report · Acceptance letter]

22 Feb 2024

PONE-D-23-19599R2 

PLOS ONE

Dear Dr. Hirosawa, 

I'm pleased to inform you that your manuscript has been deemed suitable for publication in PLOS ONE. Congratulations! Your manuscript is now being handed over to our production team.

Kind regards, 

on behalf of

Dr. Thiago P. Fernandes 

Academic Editor

PLOS ONE